# The Adaptive Complexity of Minimizing Relative Fisher Information

**Huanjian Zhou**
Graduate School of Frontier Sciences
The University of Tokyo
Center for Advanced Intelligence Project
RIKEN
zhou@ms.k.u-tokyo.ac.jp

**Masashi Sugiyama**
Center for Advanced Intelligence Project
RIKEN
Graduate School of Frontier Sciences
The University of Tokyo
sugi@k.u-tokyo.ac.jp

## Abstract

Non-log-concave sampling from an unnormalized density is fundamental in machine learning and statistics. As datasets grow larger, computational efficiency becomes increasingly important, particularly in reducing adaptive complexity, namely the number of sequential rounds required for sampling algorithms. In this work, we initiate the study of the adaptive complexity of non-log-concave sampling within the framework of relative Fisher information introduced by Balasubramanian et al. in 2022. To obtain a relative Fisher information of at most $\varepsilon^2$ from the target distribution, we propose a novel algorithm that reduces the adaptive complexity from $\mathcal{O}(d^2/\varepsilon^4)$ to $\mathcal{O}(d/\varepsilon^2)$ by leveraging parallelism. Furthermore, we show our algorithm is optimal for a specific regime of large $\varepsilon$. Our algorithm builds on a diagonally parallelized Picard iteration, while the lower bound is based on a reduction from the problem of finding stationary points.

## 1 Introduction

We study the problem of adaptive sampling from a target distribution over $\mathbb{R}^d$ given query access to its unnormalized density, a fundamental task in areas such as Bayesian inference, randomized algorithms, and machine learning [MR+07, NWS19, RCC99]. Recently, significant progress has been made in developing sequential algorithms for this problem, drawing inspiration from the extensive optimization toolkit, particularly when the target distribution is log-concave [JKO98, DMM19, MCC+21]. Typically, when access to the function value is available, many high-accuracy samplers[1] have been designed based on Metropolis–Hastings filters or a proximal sampler [DCWY19, CDWY20, LST20, ALPW24, LST21, AC24, FYC23].

However, in many practical applications, such as energy-based models and Markov Decision Processes, evaluating the log-likelihood is often computationally intractable [LCH+06, SB13]. In such scenarios, an alternative approach to designing high-accuracy samplers involves leveraging parallelism [SL19, YD24, ACV24, ZS24]. These algorithms effectively leverage contemporary parallel computing resources, such as multi-core central processing units (CPUs) and many-core graphics processing units (GPUs), especially since log-likelihood gradient evaluations often admit parallelization [HLB+21, HLFS21]. In particular, the authors [SL19, YD24, ACV24, ZS24] proposed samplers that find an $\varepsilon$-accurate solution within $\mathcal{O}(\text{poly}\log(d/\varepsilon^2))$ iterations, significantly improving upon the sub-polynomial complexity of $\mathcal{O}(d^a/\varepsilon^b)$ for log-concave distributions for some constant $a, b \in (0, 1)$.

39th Conference on Neural Information Processing Systems (NeurIPS 2025).

---

[1]Samplers with complexity $\text{poly}(d, \log(K_0/\varepsilon))$, assuming constant smoothness and condition number and $K_0$ as initial KL divergence.

In contrast, there are comparatively few works which study the adaptive complexity[2] when the target distribution is not strongly log-concave or are multimodal, such as mixtures of Gaussians.

To study the complexity of sampling from non-log-concave distributions, a general framework inspired by stationary point analysis in non-convex optimization (see, e.g., [N+18]) has been developed. Specifically, Balasubramanian et al. [BCE+22] proposed defining an $\varepsilon$-stationary point for sampling as any measure $\mu$ satisfying $\sqrt{\mathsf{FI}(\mu\|\pi)} \leq \varepsilon$, where $\mathsf{FI}(\mu\|\pi) = \mathbb{E}_\mu\left[\left\|\nabla \log(\mu/\pi)\right\|^2\right]$ denotes the *relative Fisher information* between $\mu$ and the target distribution $\pi$. They demonstrated that averaged Langevin Monte Carlo finds an $\varepsilon$-stationary point within $\mathcal{O}(dK_0/\varepsilon^4)$ iterations, where $K_0 := \mathsf{KL}(\mu_0\|\pi)$ represents the initial Kullback–Leibler divergence from the initial measure $\mu_0$ to $\pi$. Furthermore, Chewi et al. [CGLL23] established an $\Omega(1/\varepsilon^2)$ query complexity lower bound for this setting. Existing studies offer only a few bounds on the query complexity of non-log-concave sampling, and our understanding of the adaptive complexity remains critically limited. This gap motivates our investigation into the question:

> *How many sequential rounds are needed to minimize* $\mathsf{FI}$ *for non-log-concave sampling?*

## 1.1 Our Contribution

In this paper, we establish the *first* upper and lower bounds for the parallel runtime complexity of sampling. We now informally describe our main results. We assume access to an initial point $x^0 \sim \mu_0$, where the KL divergence from the target distribution is $K_0 = \mathsf{KL}(\mu_0\|\pi)$ and the target distribution $\pi$ is $L$-log-smooth.

**New parallelized algorithm with improved complexity.** By parallelizing the averaged Langevin Monte Carlo [BCE+22], which has optimal query complexity for a specific regime of large $\mathsf{FI}$ ($\varepsilon = \sqrt{Ld}$) [CGLL23], we improve the adaptive complexity from $\mathcal{O}(\frac{L^2 dK_0}{\varepsilon^4})$ to $\mathcal{O}(\frac{LK_0}{\varepsilon^2} + \log(\frac{Ld}{\varepsilon^2}))$ (Theorem 3.1). When all parameters except the dimension $d$ are treated as constants, our algorithm improves the adaptive complexity from $\mathcal{O}(d)$ to $\mathcal{O}(\log d)$, matching the parallel speedup known for strongly log-concave sampling [ZS24]. Moreover when $K_0 = \mathcal{O}(d)$, which is common assumption and the analog of the optimal gap ($f(0) - \min f(x) \lesssim d$) in non-convex optimization [CEL+24, Appendix A], our algorithm achieves an adaptive complexity of $\widetilde{\mathcal{O}}(\frac{Ld}{\varepsilon^2})$, improving over the prior complexity of $\mathcal{O}(\frac{L^2 d^2}{\varepsilon^4})$.

**Lower bound.** We further prove our parallelized algorithm is optimal for a specific regime of large $\mathsf{FI}$ ($\varepsilon = \sqrt{Ld}$) by showing when $\widetilde{\mathcal{O}}(K_0) \geq d \geq \widetilde{\Omega}(K_0^{2/3})$, the adaptive complexity is $\Omega(\frac{K_0}{d})$ (Theorem 4.1). For the accuracy level $\varepsilon = \sqrt{Ld}$, the adaptive complexity of the parallelized averaged Langevin Monte Carlo matches that of its sequential counterpart. Therefore, the sequential algorithm proposed by Balasubramanian et al. [BCE+22] is also adaptively optimal in this regime. We summarize the comparison between existing bounds and our results in Table 1.

Moreover, although this lower bound only applies to a specific accuracy regime, it rules out the possibility of a general high-accuracy sampler via parallelism. This highlights a stark separation between log-concave and non-log-concave sampling, whereas Zhou et al.[ZS24] developed a general high-accuracy sampler via parallelism for strongly log-concave distributions, along with a tight lower bound on the accuracy [ZWS24].

**A separation between optimization and sampling.** Our work also highlights a fundamental difference between sampling and optimization: Unlike in sampling, no analogous separation exists between convex and non-convex optimization, as parallelism fails to accelerate gradient descent for either class in high-dimensional settings [BS18b, DG19, ZHTS25]. In contrast, for sampling, parallelism can accelerate Langevin Monte Carlo or its averaged version for both the strongly log-concave case [ZS24] and non-log-concave case (Theorem 3.1).

---

[2]Adaptive complexity refers to the minimal number of sequential rounds required for an algorithm to achieve a desired accuracy, assuming polynomially many queries can be executed in parallel at each round [BS18a, ZWS24].

Table 1: Comparisons of our lower bounds and upper bounds. Here, $\widetilde{\Omega}$ and $\widetilde{\mathcal{O}}$ omit logarithmic factors. $K_0$ denotes the initial KL divergence, defined as $K_0 = \mathsf{KL}(\mu_0 \| \pi)$, where the initial point is drawn from the distribution $\mu_0$.

| Works | Adaptive Complexity | Queries per Iteration |
|---|---|---|
| Sequential averaged Langevin Monte Carlo [BCE$^+$22, Theorem 2] | $\mathcal{O}(\frac{L^2 d K_0}{\varepsilon^4})$ | 1 |
| Parallelized averaged Langevin Monte Carlo Theorem 3.1 | $\mathcal{O}(\frac{L K_0}{\varepsilon^2} + \log(\frac{Ld}{\varepsilon^2}))$ | $\widetilde{\mathcal{O}}(\frac{L^2 d K_0}{\varepsilon^4})$ |
| Lower bound for $\varepsilon = \sqrt{Ld}$ [CGLL23, Theorem 9] | $\Omega(\frac{K_0}{d})$ | 1 |
| Lower bound for $\varepsilon = \sqrt{Ld}$ Theorem 4.1 | $\Omega(\frac{K_0}{d})$ | $\mathsf{poly}(d)$ |

## 1.2 Related works

**Related works for minimizing** FI. The relative Fisher information (FI) between two distributions quantifies the score matching error, which is the expected squared distance between their score functions (the gradients of their log-densities). In contrast, KL divergence (KL) compares the ratio of their density functions. When the reference distribution satisfies a log-Sobolev inequality or Poincaré inequality, small FI implies small KL or small total variation (TV). However, it is possible for KL or TV to remain small even when FI becomes arbitrarily large (see [Wib25, Appendix D]). Moreover, when the reference distribution is not log-concave, FI can be made arbitrarily small while TV remains bounded away from zero [BCE$^+$22, Proposition 1].

A line of works in the literature investigates the mixing of FI for the sequential methods. For non-log-concave case, Balasubramanian et al. [BCE$^+$22] analyzed the averaged Langevin Monte Carlo and Chewi et al. [CGLL23] proved two query complexity lower bound. For log-concave case, Chewi et al. [CGLL23, Appendix A] established a high-accuracy mixing time for proximal sampler with post-processing via the heat flow. For strongly log-concave case, Wibisono [Wib25] proved exponential convergence of the proximal sampler using the strong data processing inequality.

**Related works for minimizing** TV **or** KL **for non-log-concave case.** Another line of works study the complexity of minimizing TV or KL for non-log-concave sampling. For mixture Gaussian distributions with components sharing the same shape, Ge et al. [LRG18] showed polynomially many queries are sufficient to minimize TV for simulated tempering Langevin Monte Carlo. For general non-log-concave distributions, Guo et al. [GTC24] analyzed annealed Langevin Monte Carlo and established polynomial query complexity in terms of the action associated with a curve of probability measures interpolating the target distribution and a readily sampleable distribution. Notably, when the components of a mixture of Gaussians share the same shape and mode norm, the corresponding action grows only polynomially with respect to the dimension. Recently, He et al. [HZ25] proved that the optimal query complexity scales as $(\frac{L \mathsf{m}_2}{d\varepsilon})^{\Theta(d)}$ where $\mathsf{m}_2$ is the second moment of the target distribution.

## 2 Preliminaries

### 2.1 Problem setting

Given the potential function $V : \mathcal{D} \to \mathbb{R}$, the goal of the sampling task is to draw a sample from the density $\pi_V = Z_V^{-1} \exp(-V)$, where $Z_V := \int_{\mathcal{D}} \exp(-V) \mathrm{d}\mathbf{x}$ is the normalizing constant.

**Distribution class and assumption.** If $V$ is twice-differentiable and $\nabla^2 V \preceq LI$ with $L > 0$ (where $\preceq$ denotes the Loewner order and $I$ is the identity matrix), we say the distribution $\pi_V$ is *L-log-smooth*.

**Oracle.** Given the potential function $V$, and a query $\boldsymbol{x} \in \mathcal{D}$, the 0-th order oracle answers the function value $V(\boldsymbol{x})$ and the 1-st order oracle answers both $V(\boldsymbol{x})$ and its gradient value $\nabla V(\boldsymbol{x})$. We

denote the oracle as Or. We also extend the oracle to parallel case, with input as $\{\boldsymbol{x}_1, \ldots, \boldsymbol{x}_k\} \in \mathbb{R}^{dk}$ and return multiple answers $\{V(\boldsymbol{x}_1), \nabla V(\boldsymbol{x}_1), \ldots, V(\boldsymbol{x}_k), \nabla V(\boldsymbol{x}_k)\} \in \mathbb{R}^{2dk}$ with $k = \mathsf{poly}(d)$.

**The adaptive algorithm class** The class of *adaptive* algorithms is formally defined as follows [DG19]. For any dimension $d$, an adaptive algorithm A takes $V : \mathbb{R}^d \to \mathbb{R}$ and a (possibly random) initial point $\boldsymbol{x}^0$ and iteration number $r$ as input and returns an output $\boldsymbol{x}^{r+1}$, which is denoted as $\mathsf{A}[V, \boldsymbol{x}^0, r] = \boldsymbol{x}^{r+1}$. At iteration $i \in [r] := \{1, \ldots, r\}$, A performs a batch of queries

$$Q^i = \{\boldsymbol{x}^{i,1}, \ldots, \boldsymbol{x}^{i,k_i}\}, \quad \text{with } \boldsymbol{x}^{i,j} \in \mathcal{D}, \ j \in [k_i], \ k_i = \mathsf{poly}(d),$$

such that for any $m, n \in [k_i]$, $\boldsymbol{x}^{i,m}$ and $\boldsymbol{x}^{i,n}$ are *conditionally independent* given all existing queries $\{Q^j\}_{j \in [i-1]}$ and $\boldsymbol{x}^0$. Give queries set $Q_i$, the oracle returns a batch of answers: $\mathsf{Or}(Q_i) = \{\mathsf{Or}\boldsymbol{x}^{i,1}), \ldots, \mathsf{Or}(\boldsymbol{x}^{i,k_i})\}$.

An adaptive algorithm A is *deterministic* if in every iteration $i \in \{0, \ldots, r\}$, A operates with the form

$$Q^{i+1} = \mathsf{A}^i(Q^0, \mathsf{Or}(Q^0), \ldots, Q^i, \mathsf{Or}(Q^i)),$$

where $\mathsf{A}^i$ is mapping into $\mathbb{R}^{dk_{i+1}}$ with $Q^{r+1} = \boldsymbol{x}^{r+1}$ as output and $Q^0 = \boldsymbol{x}^0$ as an initial point. We denote the class of adaptive deterministic algorithms by $\mathcal{A}_{\mathrm{det}}$.

An adaptive *randomized* algorithm has the form

$$Q^{i+1} = \mathsf{A}^i(\xi_i, Q^0, \mathsf{Or}(Q^0), \ldots, Q^i, \mathsf{Or}(Q^i)),$$

given access to a random uniform variable on $[0, 1]$ (i.e., infinitely many random bits), where $\mathsf{A}^i$ is mapping into $\mathbb{R}^{dk_{i+1}}$. We denote the class of adaptive randomized algorithms by $\mathcal{A}_{\mathrm{rand}}$.

**Measure of the output** Consider the joint distribution of all involved points $\{\mathbf{x} : \mathbf{x} \in Q^i, i = 0, \ldots, r+1\}$ and the random bits $\xi_i$. Let the marginal distribution of the output $\mathbf{x}^{r+1}$ be $\rho$. We say the output to be $\varepsilon^2$-accurate in relative Fisher information (FI) if $\mathsf{FI}(\rho, \pi_V) := \mathbb{E}_\rho \left[ \|\nabla \log(\rho/\pi)\|^2 \right] \leq \varepsilon^2$.

**Initialization.** We assume access to an initialization oracle that returns a sample from a distribution $\mu_0$ satisfying $\mathsf{KL}(\mu_0 \| \pi) \leq K_0$ since it suffice to find a stationary point which lies in a ball of radius $\mathcal{O}(\sqrt{d})$, centered at the minimizer of $f$ [CEL$^+$24]. And such a stationary point can be found fast in both strongly-convex or non-convex cases [BV04, BM20].

**Notion of complexity** Given $\varepsilon > 0$, $V \in \mathcal{F}$, and some algorithm A, define the running iteration $\mathsf{T}(\mathsf{A}, V, K_0, \varepsilon)$ as the minimum number of rounds such that given a initial point with initial KL divergence upper bounded as $K_0$, algorithm A outputs a solution $\boldsymbol{x}$ whose marginal distribution $\rho$ satisfies $\mathsf{FI}(\rho, \pi_V) \leq \varepsilon$, i.e., $\mathsf{T}(\mathsf{A}, V, K_0, \varepsilon) = \sup \{\boldsymbol{x}^0 \sim \rho_0, \ s.t. \ \mathsf{KL}(\rho_0 \| \pi) \leq K_0 : \inf \{t : \mathsf{FI}(\rho(\mathsf{A}[V, \boldsymbol{x}^0, t]), \pi_f) \leq \varepsilon\}\}^3$. We define the *worst case* complexity as

$$\mathsf{Comp}_{\mathsf{WC}}(\mathcal{F}, \varepsilon, K_0) := \inf_{\mathsf{A} \in \mathcal{A}_{\mathrm{det}}} \sup_{V \in \mathcal{F}} \mathsf{T}(\mathsf{A}, V, K_0, \varepsilon).$$

For some randomized algorithm $\mathsf{A} \in \mathcal{A}_{\mathrm{rand}}$, we define the *randomized* complexity as[4]

$$\mathsf{Comp}_{\mathsf{R}}(\mathcal{F}, \varepsilon, K_0) := \inf_{\mathsf{A} \in \mathcal{A}_{\mathrm{rand}}} \sup_{V \in \mathcal{F}} \mathsf{T}(\mathsf{A}, V, K_0, \varepsilon).$$

By definition, we have $\mathsf{Comp}_{\mathsf{WC}}(\mathcal{F}, \varepsilon, K_0) \geq \mathsf{Comp}_{\mathsf{R}}(\mathcal{F}, \varepsilon, K_0)$. In the rest of this paper, we only consider the randomized complexity and we lower-bound it by considering the *distributional* complexity:

$$\mathsf{Comp}_{\mathsf{D}}(\mathcal{F}, \varepsilon, K_0) := \sup_{F \in \Delta(\mathcal{F})} \inf_{\mathsf{A} \in \mathcal{A}_{\mathrm{rand}}} \mathbb{E}_{V \sim F} \mathsf{T}(\mathsf{A}, V, K_0, \varepsilon),$$

where $\Delta(\mathcal{F})$ is the set of probability distributions over the class of functions $\mathcal{F}$.

---

[3] We note that in sampling, the iteration complexity is determined by the output of the last iteration, which is analogous to last-iteration properties in optimizations [ALW19].

[4] We note that in sampling, we cannot define the randomized complexity as the expected running iteration over mixtures of deterministic algorithms as in the case of optimization [BGP17], since the intrinsic randomness $\xi_i$ will affect the marginal distribution of output. Furthermore, Yao's minimax principle [AB09] cannot be applied, since the different definition of randomized complexity. We acknowledge that another possible option not discussed in this paper is the "Las Vegas" algorithm, which can return "failure," as described in [AC24].

## 2.2 Averaged Langevin Monte Carlo

One of the most commonly-used dynamics for sampling is Langevin dynamics [Che23], which is the solution to the following SDE, $\mathrm{d}\boldsymbol{x} = -\nabla V(\boldsymbol{x})\mathrm{d}t + \sqrt{2}\mathrm{d}\boldsymbol{B}_t$, where $(\boldsymbol{B}_t)_{t\in[0,T]}$ is a standard Brownian motion in $\mathbb{R}^d$. When $\pi$ is $\alpha$ strongly log-concave or $V$ is $\alpha$ strongly convex, the time derivative of the relative Fisher information satisfies

$$\partial_t\mathsf{FI}(\mu_t\|\pi) \leq -2\alpha\mathsf{FI}(\mu_t\|\pi),$$

where $\mu_t$ is the law at time $t$, (see Section 2.5 [Wib25]). To the best of our knowledge, when $\pi_V$ is non-log-concave, its contraction properties remain unknown. However, a discrete-time analog of the de Bruijn identity holds for the Langevin Monte Carlo with step size $h \lesssim \frac{1}{L}$ [BCE+22, Appendix B]:

$$\partial_t\mathsf{KL}(\mu_t\|\pi) \leq -\frac{1}{2}\mathsf{FI}(\mu_t\|\pi) + \mathcal{O}(L^2 dh).$$

By integrating and summing, the averaged $\mathsf{FI}$ along Langevin Monte Carlo can be bounded as

$$\frac{1}{Nh}\int_0^{Nh}\mathsf{FI}(\mu_t\|\pi)\mathrm{d}t \leq \frac{2\mathsf{KL}(\mu_0\|\pi)}{Nh} + \mathcal{O}(L^2 dh).$$

By the convexity of the Fisher information, it is sufficient to output a sample from the averaged distribution $\bar{\mu}_{Nh} = \frac{1}{Nh}\int_0^{Nh}\mu_t\mathrm{d}t$.

## 2.3 Parallelized Langevin Monte Carlo

The main idea of parallelized Langevin Monte Carlo is to regroup the discrete grids along time horizon and update all grids in same group simultaneously [SL19, ACV24, YD24, ZS24]. Specifically, taking Picard iteration [Cle57, ACV24] as example, to approximate the difference $\boldsymbol{x}_{t_{n+1}} - \boldsymbol{x}_{t_n}$ over time slice $[t_n, t_{n+1}]$ as

$$\boldsymbol{x}_{t_{n+1}} - \boldsymbol{x}_{t_n} = \int_{t_n}^{t_{n+1}} V(\boldsymbol{x}_s)\mathrm{d}s + \sqrt{2}(\boldsymbol{B}_{t_{n+1}} - \boldsymbol{B}_{t_n})$$

$$\approx \sum_{i=1}^{M} w_i V(\boldsymbol{x}_{t_n+\tau_{n,i}})\mathrm{d}s + \sqrt{2}(\boldsymbol{B}_{t_{n+1}} - \boldsymbol{B}_{t_n}),$$

with a discrete grid of $M$ collocation points as $t_n = t_n + \tau_{n,0} \leq t_n + \tau_{n,1} \leq t_n + \tau_{n,2} \leq \cdots \leq t_n + \tau_{n,M} = t_{n+1}$. We update the points in a wave-like fashion, which inherently allows for parallelization: for $m' = 1, \ldots, M$, $p = 0, 1, \ldots, K - 1$,

$$\boldsymbol{x}_{t_n+\tau_{n,m}}^{p+1} = \boldsymbol{x}_{t,n} + \sum_{m=1}^{M-1} w_m V(\boldsymbol{x}_{t_n+\tau_{n,m}}^p) + \sqrt{2}(\boldsymbol{B}_{t_n+\tau_{n,m}} - \boldsymbol{B}_{t_n}).$$

With such regrouping, as long as the total time length of each group scales as $\mathcal{O}(1/L)$, the grids will converge exponentially fast. Given a sufficiently accurate starting point at time $t_n$, the initial error scales as $\mathcal{O}(d)$. Therefore, $K = \mathcal{O}\left(\log\left(\frac{d}{\varepsilon^2}\right)\right)$ steps suffice for the convergence of each group. In the strongly log-concave case, it suffices to simulate Langevin dynamics over the time interval $[0, \mathcal{O}(\log(\frac{\mathsf{KL}(\mu_0\|\pi)}{\varepsilon^2}))]$. Therefore, $N = \mathcal{O}(\log(\frac{\mathsf{KL}(\mu_0\|\pi)}{\varepsilon^2}))$ groups are sufficient, and the total number of steps scales as $KN = \mathcal{O}(\log^2(\frac{d}{\varepsilon^2}))$, assuming $\mathsf{KL}(\mu_0\|\pi) = \mathcal{O}(d)$. Recently, Zhou et al. [ZS24] showed the sequential update over each group is not necessary and proposed a diagonal style update with $\mathcal{O}(\log(\frac{d}{\varepsilon^2}))$ total steps (See Figure 1).

## 3 Parallel Picard method for minimizing relative Fisher information

In this section, we present parallel Picard methods for minimizing relative Fisher information in non-log-concave case (Algorithm 1) and show it holds improved convergence rate (Theorem 3.1). We illustrate the algorithm in Section 3.1, and give a proof sketch in Section 3.2. All the missing proofs can be found in Appendix A.

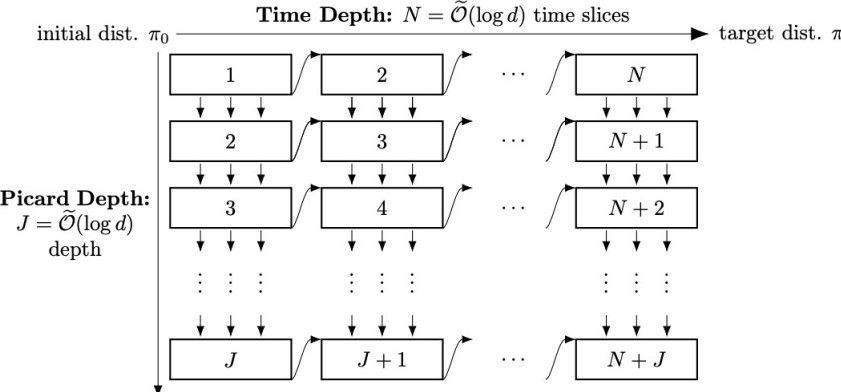

Figure 1: Illustration of the parallel Picard method: each rectangle represents an update, and the number within each rectangle indicates the index of the Picard iteration. The approximate time complexity is $N + J = \widetilde{\mathcal{O}}(\log d)$.

**Theorem 3.1.** *Given two integer parameters $J \geq N$ and $M$ and an access to the gradient oracle of $\nabla V$, there is an algorithm that runs $N + J$ iterations with at most $M(N + J)$ queries per iteration and outputs a sample with marginal distribution $\rho$ such that*

$$\mathsf{FI}(\rho\|\pi) \lesssim \underbrace{\frac{L\mathsf{KL}(\mu_0\|\pi)}{N}}_{\text{convergence of averaged LMC}} + \underbrace{\frac{Ld}{M}}_{\text{discretization error}} + \underbrace{\left(\frac{L\mathsf{KL}(\mu_0\|\pi)}{N} + Ld\right)0.5^{J-N}}_{\text{parallization error}}.$$

(1)

*Furthermore, by setting $N = \frac{L\mathsf{KL}(\mu_0\|\pi)}{\varepsilon^2}$ and $M = \frac{Ld}{\varepsilon^2}$ and $J - N = \mathcal{O}(\log(\frac{Ld}{\varepsilon^2}))$, the algorithm runs within $\mathcal{O}(\frac{LK_0}{\varepsilon^2} + \log\left(\frac{Ld}{\varepsilon^2}\right))$ iterations with at most $\mathcal{O}(\frac{L^2dK_0}{\varepsilon^4} + \frac{Ld}{\varepsilon}\log\left(\frac{Ld}{\varepsilon^2}\right))$ queries per iteration, and returns a $\varepsilon^2$-accurate sample $\boldsymbol{x}$ in $\mathsf{FI}$.*

**Remark 3.2.** *We can usually take $\mathsf{KL}(\mu_0\|\pi) = \mathcal{O}(d)$. Then taking*

$$N = \mathcal{O}\left(\frac{Ld}{\varepsilon^2}\right), \quad M = \mathcal{O}\left(\frac{Ld}{\varepsilon^2}\right), J = N + \mathcal{O}\left(\log\frac{Ld}{\varepsilon^2}\right).$$

*the algorithm runs $\mathcal{O}\left(\frac{Ld}{\varepsilon^2}\right)$ iterations with $\mathcal{O}\left(\frac{L^2d^2}{\varepsilon^4}\right)$ queries per iteration and return a sample having $\varepsilon$ accuracy in terms of the Fisher information w.r.t. the target.*

**Remark 3.3.** *Compared to the bound for the sequential method presented in [BCE$^+$22], our upper bound (right-hand side of Eq. (1)) for the parallel method also include one converge term $\frac{L\mathsf{KL}(\mu_0\|\pi)}{N}$ and a discretization error term $\frac{Ld}{M}$, and an additional exponentially decaying error term by parallelism $\left(\frac{L\mathsf{KL}(\mu_0\|\pi)}{N} + Ld\right)0.5^{J-N}$.*

**Remark 3.4** (**Tradeoff between query per round and adaptive complexity**). *When the number of computation cores, denoted by $W$, is limited, the adaptive complexity of our algorithm is $\widetilde{\mathcal{O}}\left(\frac{d}{\varepsilon^2} + \frac{d^2}{\varepsilon^4W}\log\left(\frac{d}{\varepsilon^2}\right)\right)$. When $W = 1$, this recovers the sequential method; when $W = \widetilde{\mathcal{O}}\left(\frac{d^2}{\varepsilon^4}\right)$ with assumption $K_0 = \mathcal{O}(d)$, it recovers our fully parallel method. Another interesting intermediate case arises when applying averaged Langevin Monte Carlo to the algorithm in [ACV24], which updates time slices sequentially. This corresponds to $W = \frac{d}{\varepsilon^2}$, yielding an adaptive complexity of $\mathcal{O}\left(\frac{d}{\varepsilon^2}\log(\frac{d}{\varepsilon^2})\right)$ which fits naturally within the above tradeoff curve.*

## 3.1 Parallel Picard method

We adopt the parallel Picard method [ZS24] which achieve nearly tight result for log-concave sampling. In Lines 5–9, we apply the averaged Langevin Monte Carlo [BCE$^+$22] to initialize the vector value at all grids with Fisher information bounded by $\mathcal{O}(d)$. Specifically, we initialize $\boldsymbol{x}_{n,m}^0$ at all points along the time horizon using the output of the averaged Langevin Monte Carlo procedure

in Line 9. In Lines 1–4, we generate the random noises and fixed them. Subsequently, we seek to construct an approximate path of the true Langevin dynamics by means of parallel computation. Specifically, in Lines 10–16, we apply parallel Picard method with forward Euler-Maruyama Method in diagonal style as illustrated in Figure 1. In $j$-th each update, for $m$-th grid in $n$-th time slices, we perform

$$\boldsymbol{x}_{n,m}^j = \boldsymbol{x}_{n,0}^j - \frac{h}{M}\sum_{m'=0}^{m-1}\nabla V(\boldsymbol{x}_{n,m'}^{j-1}) + \sqrt{2}(B_{nh+mh/M} - B_{nh}),$$

where $\boldsymbol{x}_{n,m}^{\cdot}$ corresponds to time $nh + \frac{m}{M}h$. In Lines 17, we return the average point along the interpolation path of $\{\boldsymbol{x}_{n,m}^j : n \in [N], m \in [M]\}$.

---

**Algorithm 1:** Parallel Picard method for non-log-concave sampling

---

**Input :** $\boldsymbol{x}^0 \sim \mu_0$, gradient oracle of $\nabla V$, the number of the iterations in outer loop $J$, the number of time slices $N$, the length of time slices $h$, the number of points on each time slices $M$.

1 **for** $n = 0, \dots, N-1$, $m = 0, \dots, M$ *(in parallel)* **do**
2     $\xi_{n,m} = \mathcal{N}(0, (h/M)\boldsymbol{I}_d)$                    ▷ `generate the noise`

3 **for** $n = 0, \dots, N-1$, $m = 0, \dots, M$ *(in parallel)* **do**
4     $B_{nh+(m+1)h/M} = \sum\limits_{n'=\{0,\dots,n-1\}}\sum\limits_{m'\in[M]}\xi_{n',m'} + \sum\limits_{m'\in[m]}\xi_{n,m'}$

5 $\boldsymbol{x}_{-1,0}^j = \boldsymbol{x}^0$, for $j = -1, \dots, J$
6 **for** $k = 1, \dots, N$ **do**
7     $\boldsymbol{x}_{kh}^{-1} = \boldsymbol{x}_{(k-1)h}^{-1} - h\nabla V(\boldsymbol{x}_{(k-1)h}^{-1}) + \sqrt{2}(B_{kh} - B_{(k-1)h})$,      ▷ `initialization`

8 Pick a time $t \in [0, Nh]$ uniformly at random,
9 Let $k$ be the largest integer such that $kh \le t$, for all $n = 0, \dots, N-1$ and $m \in [M]$, set

$$\boldsymbol{x}_{n,m}^0 = \boldsymbol{x}_{kh}^{-1} - (t-kh)\nabla V(\boldsymbol{x}_{kh}^{-1}) + \sqrt{2}(B_t - B_{kh}).$$

    **for** $k = 1, \dots, N$ **do**
10      **for** $j = 1, \dots, \min\{k-1, J\}$ *and* $m = 1, \dots, M$ *(in parallel)* **do**
11         let $n = k - j$ and $\boldsymbol{x}_{n,0}^j = \boldsymbol{x}_{n-1,M}^j$,
12         $\boldsymbol{x}_{n,m}^j = \boldsymbol{x}_{n,0}^j - \frac{h}{M}\sum\limits_{m'=0}^{m-1}\nabla V(\boldsymbol{x}_{n,m'}^{j-1}) + \sqrt{2}(B_{nh+mh/M} - B_{nh})$,

13 **for** $k = N+1, \dots, N+J-1$ **do**
14      **for** $n = \max\{0, k-J\}, \dots, N-1$ *and* $m = 1, \dots, M$ *(in parallel)* **do**
15         let $j = k - n$ and $\boldsymbol{x}_{n,0}^j = \boldsymbol{x}_{n-1,M}^j$,
16         $\boldsymbol{x}_{n,m}^j = \boldsymbol{x}_{n,0}^j - \frac{h}{M}\sum\limits_{m'=0}^{m-1}\nabla V(\boldsymbol{x}_{n,m'}^{j-1}) + \sqrt{2}(B_{nh+mh/M} - B_{nh})$,

17 Pick a time $t \in [0, Nh]$ uniformly at random, let $k$ be the largest integer such that $kh/M \le t$,

$$\boldsymbol{x}_t = \boldsymbol{x}_{\lfloor k/M \rfloor, k-\lfloor k/M \rfloor M}^J - (t - kh/M)\nabla V(\boldsymbol{x}_{\lfloor k/M \rfloor, k-\lfloor k/M \rfloor M}^J) + \sqrt{2}(B_t - B_{\lfloor kh/M \rfloor}).$$

**return** $\boldsymbol{x}_t$.

---

### 3.2 Proof sketch of Theorem 3.1: analysis of Algorithm 1

Following [VW19, ACV24, ZS24], we use interpolation method to obtain a discrete-time analog of de Bruijn identity:

$$\partial_t \mathsf{KL}(\mu_t \| \pi) \le -\frac{3}{4}\mathsf{FI}(\mu_t \| \pi) + \mathbb{E}\left[\left\|\nabla V(\boldsymbol{x}_t) - \nabla V(\boldsymbol{x}_{n,m}^{J-1})\right\|^2\right].$$

By Lipschitz condition, Integrating over time $t \in \left[nh + \frac{mh}{M}, nh + \frac{(m+1)h}{M}\right]$ and summing, we can upper bound the averaged $\mathsf{FI}$ along the discrete trajectory by

$$\frac{1}{Nh}\int_0^{Nh}\mathsf{FI}(\mu_t\|\pi)\mathrm{d}t \le \frac{2K_0}{Nh} + \frac{2Ld}{M} + 3L^2\mathcal{E},$$

where $\mathcal{E} = \max\limits_{n \in [N], \, m \in [M]} \mathbb{E}\left[\left\|\boldsymbol{x}_{n,m}^J - \boldsymbol{x}_{n,m}^{J-1}\right\|^2\right]$ which represents the convergence error of Picard iteration. It remains to prove the convergence of Picard iteration. The key is to decompose the error during the diagonal update and upper bound the initial error. Let $\mathcal{E}_n^j := \max\limits_{m \in [M]} \mathbb{E}\left[\left\|\boldsymbol{x}_{n,m}^j - \boldsymbol{x}_{n,m}^{j-1}\right\|^2\right]$. By definition of Euler-Maruyama scheme, we can decompose the error as

$$\mathcal{E}_n^j \leq 2\mathcal{E}_{n-1}^j + 2h^2 L^2 \mathcal{E}_{n-1}^j,$$

where $h$ is the length of time slices, and $L$ is Lispchitz constant of the gradient. Thus by choosing the time length $h$ sufficiently small relative to the Lipschitz constant $L$, we ensure convergence along the Picard direction. As for the initial error, we decompose it as

$$\mathcal{E}_n^1 \leq 2\mathcal{E}_{n-1}^1 + 2h^2 \mathsf{FI}(\mu^0 \| \pi) + 5dh,$$

where $\mathsf{FI}(\mu^0 \| \pi)$ is controlled by the initialization procedure using averaged Langevin Monte Carlo with a large step size $h = \mathcal{O}(1)$.

The details of the proof can be found in Appendix A.

## 4  Lower bound

In this section, we establish our lower bound (Theorem 4.1) by combining a reduction from minimizing the relative Fisher information to the problem of finding a stationary point with the optimal adaptive complexity for this task, as established by Zhou et al. [ZHTS25].

**Theorem 4.1.** *Let the dimension $d$ satisfies $\widetilde{\mathcal{O}}(K_0) \geq d \geq \widetilde{\Omega}(K_0^{2/3})$. There exists a function class $\mathcal{F}$, consisting of $L$-smooth functions, such that for any $\varepsilon \geq \sqrt{Ld}$ and $\boldsymbol{x}^0 \sim \rho_0$ with $\mathsf{KL}(\rho_0 \| \pi_V) \leq K_0$ for any $\{\pi_V\}_{V \in \mathcal{F}}$,*

$$\mathsf{Comp_R}(\mathcal{F}, \varepsilon, K_0) \gtrsim \frac{K_0 L}{\varepsilon^2} \gtrsim \frac{K_0}{d}.$$

**Remark 4.2.** *This lower bound matches the upper bound in Theorem 3.1 for specific regime of $\varepsilon = \sqrt{Ld}$. The condition $d \geq \widetilde{\Omega}(K_0^{2/3})$ arises because the lower bound construction in [ZHTS25] lies in high dimensional regime $d \geq \widetilde{\Omega}(\varepsilon^{-4})$.*

*Proof of Theorem 4.1.* To prove Theorem 4.1, we will reduce the problem to that of finding a stationary point in parallel, and then verify the initialization condition. We first recall the reduction lemma from non-log-concave sampling to non-convex optimization.

**Lemma 4.3** ([CEL$^+$24, Lemma 16]). *Let $\pi \propto \exp(-V)$ be a $\beta$-log-smooth density on $\mathbb{R}^d$. Then, for any probability measure $\mu$,*

$$\mathbb{E}_\mu\left[\|\nabla V\|^2\right] \leq \mathsf{FI}(\mu \| \pi) + 2\beta d.$$

To apply this reduction, we recall the adaptive complexity of finding stationary point which scales as $O(\Delta L \cdot \varepsilon^{-2})$ for high dimensional regime ($d = \widetilde{\Omega}\left(\varepsilon^{-4}\right)$).

**Theorem 4.4** (**The adaptive complexity of finding stationary points [ZHTS25]**). *Assume $d = \widetilde{\Omega}\left(\varepsilon^{-4}\right)$. There exits a function class $\mathcal{F}$ consisting of some $L$-smooth function with given initial point $\boldsymbol{x}^0$, such that $V(\boldsymbol{x}^0) - \min\limits_{\boldsymbol{x}} V(\boldsymbol{x}) \leq \Delta$, and the following holds: any (possible randomized) algorithm running within $O(\Delta L \cdot \varepsilon^{-2})$ iterations with $\mathsf{poly}(d)$ queries per iteration fails to find $\varepsilon$-approximate point for any $V \in \mathcal{F}$ with probability $1 - d^{-\omega(1)}$.*

We set $\varepsilon = 4\sqrt{Ld}$. From the reduction lemma (Lemma 4.3), if we can obtain a sample from a measure $\mu$ such that for $\pi_V \propto \exp(-V)$, it holds that $\mathsf{FI}(\mu \| \pi_V) \leq Ld$, then a sample from $\mu$ is a $\varepsilon$-stationary point of $f$ with probability at least $1/2$.

In the following, we check the initialization condition. We set initialization oracle to output a sampler from $\mu \sim \mathcal{N}(0, L^{-1} I_d)$. Now we need to compute the value of $K_0 := \sup_{V \in \mathcal{F}} \mathsf{KL}(\mu_0 \| \pi_V)$. To do so, we use the following lemma.

**Lemma 4.5** (**KL divergence at initialization [CGLL23, Lemma 17]**). *Suppose that $V : \mathbb{R}^d \mapsto \mathbb{R}$ is a function such that $V(0) - \inf V \le \Delta$, $\nabla V$ is $L$-Lipschitz, and $\mathfrak{m} := \int \|\cdot\| \, d\pi < \infty$ where $\pi \propto \exp(-V)$. Then, for $\mu_0 = \mathcal{N}(0, L^{-1}I_d)$, we have the bound*

$$\mathsf{KL}(\mu_0 \| \pi) \lesssim \Delta + d(1 \vee \log(L\mathfrak{m}^2)).$$

We remind the hardness function in [ZHTS25] takes form as

$$V(\boldsymbol{x}) = \mathsf{poly} \cdot (g(\rho(\boldsymbol{x}/\mathsf{poly}))) + \frac{1}{2\tau^2} \|\boldsymbol{x}\|^2,$$

where poly denote any positive quantity for which both the quantity and its inverse are bounded above by polynomials in $L$, $\Delta$, $d$, and $1/\varepsilon$, and $\tau = \mathsf{poly}$. Furthermore $g : \mathbb{R}^d \mapsto \mathbb{R}$ and $\rho : \mathbb{R}^d \mapsto \mathbb{R}^d$ are poly-Lipschitz, then

$$\|\nabla g(\rho(\cdot/\mathsf{poly}))\| \le \mathsf{poly}.$$

Thus $\mathsf{FI}(\pi_V \| \nu) = \mathsf{poly} \cdot \mathbb{E}_{\pi_V} [\|\| \nabla g(\rho(\cdot/\mathsf{poly}))] \le \mathsf{poly}$ where $\nu = \mathcal{N}(0, \tau I_d)$. By the Donsker–Varadhan variational principle [PW25, Theorem 4.6] and the fact that $\nu$ satisfies the log-Sobolev inequality with poly, we have

$$\begin{aligned}
\mathbb{E}_{\pi_V}\left[\|\cdot\|^2\right] &\le \frac{1}{\lambda}\left\{\mathsf{KL}(\pi_V \| \nu) + \log \mathbb{E}_\nu \exp(\lambda\|\cdot\|^2)\right\} \\
&\le \mathsf{poly}\left\{\mathsf{FI}(\pi_V \| \nu) + 1\right\} \\
&\le \mathsf{poly},
\end{aligned}$$

with $\lambda = \frac{1}{\mathsf{poly}}$ and $\mathbb{E}_\nu \exp(\lambda \| \cdot \|^2) \le 1$. Thus $K_0 = \mathsf{KL}(\mu_0 \| \pi) \lesssim \Delta + \widetilde{\mathcal{O}}(d)$. Thus if $K_0 \ge \widetilde{\Omega}(d)$, $\Delta \gtrsim K_0$. Since $\varepsilon = \sqrt{Ld}$, the number of the required iteration satisfies

$$\#\text{iteration} \gtrsim \frac{\Delta L}{\varepsilon^2} \gtrsim \frac{K_0}{d}.$$

Finally we check the requirement of dimension, $d \ge \widetilde{\Omega}(K_0/d)^2$, which is satisfied provided $d \ge \widetilde{\Omega}(K_0^{2/3})$. $\qquad\square$

## 5   Discussion and Conclusion

In this work, we initialize the studying of parallelize minimizing the relative Fisher information for non-log-concave sampling by showing (1) averaged Langevin Monte Carlo can be accelerated by parallelism and (2) offer a tight lower bound for specific accuracy regime. Our results rule out the possibility of designing general high-accuracy relative Fisher information minimizer via parallelism in the non-log-concave setting, contrasting with the log-concave case. Furthermore, our results offer a new understanding for the theme of "sampling versus optimization" by revealing the distinct role parallelism plays in separating the two.

We believe there are several intriguing directions for future work exploring the role of parallelism in sampling versus optimization, and we conclude by highlighting a few of them.

1. **(Constant-dimensional case).** In the constant-dimensional setting, it is possible to find a stationary point in $k = \mathcal{O}(\log(1/\varepsilon))$ rounds using $\mathcal{O}\left(\varepsilon^{-\frac{d-1}{2}(1+\mathcal{O}(2^{-k}))}\right)$ queries per round by leveraging gradient flow trapping [BM20, HZ23, ZHTS25]. This raises the question: can one similarly minimize the relative Fisher information within $\mathcal{O}(\log(1/\varepsilon))$ rounds by trapping the Langevin dynamics?

2. **(Lower bounds beyond specific large accuracy).** Although our lower bound is tight, it applies only to a specific high-accuracy regime ($\varepsilon = \sqrt{Ld}$). In contrast, for parallel non-convex optimization, tight lower bounds on adaptive complexity, namely, $\Omega(\varepsilon^{-2})$, are known for finding stationary points when $\varepsilon \le \widetilde{\mathcal{O}}(d^{1/4})$ [ZHTS25]. A natural question is whether similar $\mathsf{poly}(\varepsilon^{-1})$ lower bounds can be established beyond this setting, particularly in the low-accuracy regime ($\varepsilon = o(1)$), as by the bump function construction used for query complexity in Section 4 of [CGLL23].

3. **(Functional inequality case).** For strongly log-concave distributions, it is possible to design high-accuracy samplers leveraging parallelism [YD24, ACV24, ZS24]. In contrast, for non-log-concave distributions, designing general high-accuracy samplers via parallelism becomes impossible. This raises a natural question: what is the boundary between these two cases? A much weaker question is whether high-accuracy samplers can be designed via parallelism for distributions satisfying functional inequalities such as the log-Sobolev or Poincaré inequality.

## Acknowledgment

The authors thank Sinho Chewi for very helpful conversations. HZ was supported by International Graduate Program of Innovation for Intelligent World and Next Generation Artificial Intelligence Research Center. MS was supported by JST ASPIRE Grant Number JPMJAP2405.

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

# A Proof of Theorem 3.1

## A.1 Useful facts

In this section, we first recall several useful lemmas.

**Lemma A.1 (Differential inequality of KL along interpolation [BCE⁺22, Lemma 12]).** *Consider the stochastic process defined by*

$$\boldsymbol{x}_t := \boldsymbol{x}_0 - t\boldsymbol{g}_0 + \sqrt{2}\,B_t\,, \qquad for\ t \geq 0\,,$$

*where $(B_t)_{t\geq 0}$ is a standard Brownian motion in $\mathbb{R}^d$ which is independent of $(\boldsymbol{x}_0, \boldsymbol{g}_0)$. Let $\mu_t$ for the law of $\boldsymbol{x}_t$. Then*

$$\partial_t \mathsf{KL}(\mu_t \| \pi) \leq -\frac{3}{4}\mathsf{FI}(\mu_t \| \pi) + \mathbb{E}\big[\|\nabla V(\boldsymbol{x}_t) - \mathbb{E}[\boldsymbol{g}_0 \mid x_t]\|^2\big]$$

$$\leq -\frac{3}{4}\mathsf{FI}(\mu_t \| \pi) + \mathbb{E}[\|\nabla V(\boldsymbol{x}_t) - \boldsymbol{g}_0\|^2]\,.$$

**Lemma A.2 (Initialization [BCE⁺22, Theorem 2]).** *For all $n = 0, \ldots, N-1$ and $m \in [M]$, let $\mu_{n,m}^0$ be the law of $\boldsymbol{x}_{n,m}^0$, then we have*

$$\mathsf{FI}(\mu_{n,m}^0 \| \pi) \leq \frac{2\mathsf{KL}(\mu_0 \| \pi)}{Nh} + 8L^2 dh.$$

**Lemma A.3 ([CEL⁺24, Lemma 16]).** *Assume that $\nabla V$ is $L$-Lipschitz. For any probability measure $\mu$, it holds that*

$$\mathbb{E}_\mu[\|\nabla V\|^2] \leq \mathsf{FI}(\mu \| \pi) + 2dL\,.$$

## A.2 Decomposition via interpolation method

We denote $\mathsf{KL}_{n,m} = \mathsf{KL}(\mu_{n,m}^J \| \pi)$ where $\mu_{n,m}^J$ represents the law of $\boldsymbol{x}_{n,m}^J$. Let $\boldsymbol{x}_t$ be the linear interpolation between $\boldsymbol{x}_{n,m}^J$ and $\boldsymbol{x}_{n,m+1}^J$, i.e., for $t \in \left[nh + \frac{mh}{M}, nh + \frac{(m+1)hh}{M}\right]$, let

$$\boldsymbol{x}_t = \boldsymbol{x}_{n,m}^J - \left(t - nh - \frac{mh}{M}\right)\nabla V(\boldsymbol{x}_{n,m}^{J-1}) + \sqrt{2}(B_t - B_{nh+mh/M}).$$

Then Lemma A.1 yields

$$\partial_t \mathsf{KL}(\mu_t \| \pi) \leq -\frac{3}{4}\mathsf{FI}(\mu_t \| \pi) + \mathbb{E}\left[\|\nabla V(\boldsymbol{x}_t) - \nabla V(\boldsymbol{x}_{n,m}^{J-1})\|^2\right].$$

For the second term, by smooth of $V$,

$$\mathbb{E}\left[\|\nabla V(\boldsymbol{x}_t) - \nabla V(\boldsymbol{x}_{n,m}^{J-1})\|^2\right]$$

$$\leq L^2 \mathbb{E}\left[\|\boldsymbol{x}_t - \boldsymbol{x}_{n,m}^{J-1}\|^2\right]$$

$$\leq 2L^2 \mathbb{E}\left[\|\boldsymbol{x}_t - \boldsymbol{x}_{n,m}^J\|^2\right] + 2L^2 \mathbb{E}\left[\|\boldsymbol{x}_{n,m}^J - \boldsymbol{x}_{n,m}^{J-1}\|^2\right]$$

$$\leq 4L^2 \left(t - nh - \frac{mh}{M}\right)^2 \mathbb{E}\left[\|\nabla V(\boldsymbol{x}_{n,m}^{J-1})\|^2\right] + 8L^2 \mathbb{E}\left[\|B_t - B_{nh+mh/M}\|^2\right] + 2L^2 \mathbb{E}\left[\|\boldsymbol{x}_{n,m}^J - \boldsymbol{x}_{n,m}^{J-1}\|^2\right]$$

$$\leq 8L^2 \left(t - nh - \frac{mh}{M}\right)^2 \mathbb{E}\left[\|\nabla V(\boldsymbol{x}_t) - \nabla V(\boldsymbol{x}_{n,m}^{J-1})\|^2\right] + 8L^2 \left(t - nh - \frac{mh}{M}\right)^2 \mathbb{E}\left[\|\nabla V(\boldsymbol{x}_t)\|^2\right]$$

$$\quad + 8L^2 \mathbb{E}\left[\|B_t - B_{nh+mh/M}\|^2\right] + 2L^2 \mathbb{E}\left[\|\boldsymbol{x}_{n,m}^J - \boldsymbol{x}_{n,m}^{J-1}\|^2\right].$$

Taking $Lh \leq 0.1$, we have

$$\mathbb{E}\left[\|\nabla V(\boldsymbol{x}_t) - \nabla V(\boldsymbol{x}_{n,m}^{J-1})\|^2\right]$$

$$\leq 9L^2 \left(t - nh - \frac{mh}{M}\right)^2 \mathbb{E}\left[\|\nabla V(\boldsymbol{x}_t)\|^2\right] + 9L^2 \mathbb{E}\left[\|B_t - B_{nh+mh/M}\|^2\right] + 3L^2 \mathbb{E}\left[\|\boldsymbol{x}_{n,m}^J - \boldsymbol{x}_{n,m}^{J-1}\|^2\right].$$

For the first term, by Lemma A.3 and $Lh \leq 0.1$, we have

$$
\begin{aligned}
\partial_t \mathsf{KL}(\mu_t \| \pi) \leq\ & -\frac{3}{4}\mathsf{FI}(\mu_t \| \pi) + \mathbb{E}\left[\|\nabla V(\boldsymbol{x}_t) - \nabla V(\boldsymbol{x}_{n,m}^{J-1})\|^2\right] \\
\leq\ & -\left(\frac{3}{4} - \frac{9L^2 h^2}{M^2}\right)\mathsf{FI}(\mu_t\|\pi) + 18L^3 d\left(t - nh - \frac{mh}{M}\right)^2 + 9L^2 d\left(t - nh - \frac{mh}{M}\right) \\
& + 3L^2 \mathbb{E}\left[\|\boldsymbol{x}_{n,m}^J - \boldsymbol{x}_{n,m}^{J-1}\|^2\right] \\
\leq\ & -\frac{1}{2}\mathsf{FI}(\mu_t\|\pi) + 18L^3 d\left(t - nh - \frac{mh}{M}\right)^2 + 9L^2 d\left(t - nh - \frac{mh}{M}\right) \\
& + 3L^2 \mathbb{E}\left[\|\boldsymbol{x}_{n,m}^J - \boldsymbol{x}_{n,m}^{J-1}\|^2\right].
\end{aligned}
$$

Integrating it over $t \in \left[nh + \frac{mh}{M}, nh + \frac{(m+1)hh}{M}\right]$, we obtain

$$
\begin{aligned}
\mathsf{KL}_{n,m+1} - \mathsf{KL}_{n,m} \leq\ & -\frac{1}{2}\int_{nh+\frac{mh}{M}}^{nh+\frac{(m+1)hh}{M}}\mathsf{FI}(\mu_t\|\pi)\mathrm{d}t + 6L^3 d\frac{h^3}{M^3} + 5L^2 d\frac{h^2}{M^2} + 3L^2\frac{h}{M}\left[\|\boldsymbol{x}_{n,m}^J - \boldsymbol{x}_{n,m}^{J-1}\|^2\right] \\
\leq\ & -\frac{1}{2}\int_{nh+\frac{mh}{M}}^{nh+\frac{(m+1)hh}{M}}\mathsf{FI}(\mu_t\|\pi)\mathrm{d}t + 6\frac{L^2 dh^2}{M^2} + \frac{3L^2 h}{M}\mathbb{E}\left[\|\boldsymbol{x}_{n,m}^J - \boldsymbol{x}_{n,m}^{J-1}\|^2\right].
\end{aligned}
$$

Now we assume there is a uniform upper bound for $\mathbb{E}\left[\|\boldsymbol{x}_{n,m}^J - \boldsymbol{x}_{n,m}^{J-1}\|^2\right]$ for any $n \in [N]$ and $m \in [M]$, which represents the convergence error of Picard iteration. Specifically, we assume $\mathbb{E}\left[\|\boldsymbol{x}_{n,m}^J - \boldsymbol{x}_{n,m}^{J-1}\|^2\right] \leq \mathcal{E}$ for any $n \in [N]$ and $m \in [M]$. Then by summing, we have

$$\frac{1}{Nh}\int_0^{Nh}\mathsf{FI}(\mu_t\|\pi)\mathrm{d}t \leq \frac{2\mathsf{KL}_{0,0}}{Nh} + \frac{2Ld}{M} + 3L^2\mathcal{E}. \tag{2}$$

### A.3   Convergence of Picard iteration

We will end the prove by show the uniform upper bound for $\mathbb{E}\left[\|\boldsymbol{x}_{n,m}^J - \boldsymbol{x}_{n,m}^{J-1}\|^2\right]$. To bound it, we define $\mathcal{E}_n^j := \max_{m=1,\ldots,M}\mathbb{E}\left[\|\boldsymbol{x}_{n,m}^j - \boldsymbol{x}_{n,m}^{j-1}\|^2\right]$.

**Lemma A.4 (Decomposition of $\mathcal{E}_n^j$).** *Assume $Lh = \frac{1}{10}$ and let $\mu^0$ is the law of $\boldsymbol{x}_{n,0}^0$. We have the following decompositions and initialization estimations:*

1. $\mathcal{E}_n^j \leq 2\mathcal{E}_{n-1}^j + 0.02\mathcal{E}_n^{j-1}$, *for any $j = 2, \ldots, J$ and $n = 1, \ldots, N - 1$;*

2. $\mathcal{E}_n^1 \leq 2\mathcal{E}_{n-1}^1 + 2h^2\mathsf{FI}(\mu^0\|\pi) + 5dh$, *for $j = 1$ and $n = 1, \ldots, N - 1$;*

3. $\mathcal{E}_0^j \leq 0.01\mathcal{E}_0^{j-1}$, *for any $j = 2, \ldots, J$ and $n = 0$;*

4. $\mathcal{E}_0^1 \leq 2h^2\mathsf{FI}(\mu^0\|\pi) + 5dh$;

*Proof.* For $j \in [J]$, $n = 1, \ldots, N - 1$, $m = 0, \ldots, M - 1$, we have

$$
\begin{aligned}
& \mathbb{E}\left[\|\boldsymbol{x}_{n,m}^j - \boldsymbol{x}_{n,m}^{j-1}\|^2\right] \\
& \leq 2\mathbb{E}\left[\|\boldsymbol{x}_{n,0}^j - \boldsymbol{x}_{n,0}^{j-1}\|^2\right] + 2\frac{h^2}{M^2}\mathbb{E}\left[\left\|\sum_{m'=0}^{m-1}\nabla V(\boldsymbol{x}_{n,m'}^{j-1}) - \nabla V(\boldsymbol{x}_{n,m'}^{j-2})\right\|^2\right] \\
& \leq 2\mathbb{E}\left[\|\boldsymbol{x}_{n,0}^j - \boldsymbol{x}_{n,0}^{j-1}\|^2\right] + 2h^2\max_{m'=1,\ldots,m}\mathbb{E}\left[\|\nabla V(\boldsymbol{x}_{n,m'}^{j-1}) - \nabla V(\boldsymbol{x}_{n,m'}^{j-2})\|^2\right]
\end{aligned}
$$

$$\leq 2\mathcal{E}_{n-1}^{j} + 0.02\mathcal{E}_{n}^{j-1}.$$

For $j = 1$, $n = 1, \ldots, N-1$, $m = 0, \ldots, M-1$, and $p = 1, \ldots, P$, we have

$$\mathbb{E}\left[\left\|\boldsymbol{x}_{n,m}^{1} - \boldsymbol{x}_{n,m}^{0}\right\|^{2}\right]$$

$$\leq 2\mathbb{E}\left[\left\|\boldsymbol{x}_{n,0}^{1} - \boldsymbol{x}_{n,0}^{0}\right\|^{2}\right] + 2\mathbb{E}\left[\left\|\frac{h}{M}\sum_{m'=0}^{m-1}\nabla V(\boldsymbol{x}_{n,m'}^{0}) + \sqrt{2}(B_{nh+mh/M} - B_{nh})\right\|^{2}\right]$$

$$\leq 2\mathbb{E}\left[\left\|\boldsymbol{x}_{n,0}^{1} - \boldsymbol{x}_{n,0}^{0}\right\|^{2}\right] + 2h^{2}\mathbb{E}\left[\left\|\nabla V(\boldsymbol{x}_{n,m'}^{0})\right\|^{2}\right] + 4dh$$

$$\leq 2\mathbb{E}\left[\left\|\boldsymbol{x}_{n,0}^{1} - \boldsymbol{x}_{n,0}^{0}\right\|^{2}\right] + 2h^{2}(\mathsf{FI}(\mu^{0}\|\pi) + 2dL) + 4dh$$

$$\leq 2\mathcal{E}_{n-1}^{1} + 2h^{2}\mathsf{FI}(\mu^{0}\|\pi) + 5dh.$$

When $n = 0$, $j \geq 2$ we have

$$\mathbb{E}\left[\left\|\boldsymbol{x}_{0,m}^{j} - \boldsymbol{x}_{0,m}^{j-1}\right\|^{2}\right]$$

$$\leq \frac{h^{2}}{M^{2}}\mathbb{E}\left[\left\|\sum_{m'=0}^{m-1}\nabla V(\boldsymbol{x}_{0,m'}^{j-1}) - \nabla V(\boldsymbol{x}_{0,m'}^{j-2})\right\|^{2}\right]$$

$$\leq h^{2}\max_{m'=1,\ldots,m}\mathbb{E}\left[\left\|\nabla V(\boldsymbol{x}_{0,m'}^{j-1}) - \nabla V(\boldsymbol{x}_{0,m'}^{j-2})\right\|^{2}\right]$$

$$\leq L^{2}h^{2}\max_{m'=1,\ldots,m}\mathbb{E}\left[\left\|\boldsymbol{x}_{0,m'}^{j-1} - \boldsymbol{x}_{0,m'}^{j-2}\right\|^{2}\right].$$

When $n = 0$, $j = 1$ we have

$$\mathbb{E}\left[\left\|\boldsymbol{x}_{0,m}^{1} - \boldsymbol{x}_{0,m}^{0}\right\|^{2}\right] \leq 2h^{2}\mathsf{FI}(\mu^{0}\|\pi) + 5dh.$$

$\square$

**Lemma A.5.** *If $J \geq N$, then for $n = 0, \ldots, N-1$ we have*
$$\mathcal{E}_{n}^{J} \leq 200(2h^{2}\mathsf{FI}(\mu^{0}\|\pi) + 5dh)0.5^{J-N}.$$

*Proof.* By Lemma A.4, we can recursively bound $\mathcal{E}_{n}^{j}$. Specifically, for $n \geq 1$ and $j = 1$, we have
$\mathcal{E}_{n}^{1} \leq 2\mathcal{E}_{n-1}^{1} + 2h^{2}\mathsf{FI}(\mu^{0}\|\pi) + 5dh \leq 2^{n}\mathcal{E}_{0}^{1} + (2^{n}-1)(2h^{2}\mathsf{FI}(\mu^{0}\|\pi) + 5dh) \leq 2^{n+1}(2h^{2}\mathsf{FI}(\mu^{0}\|\pi) + 5dh)$,
and for $n = 0$, $j \geq 2$, $\mathcal{E}_{0}^{j} \leq 0.01^{j-1}(2h^{2}\mathsf{FI}(\mu^{0}\|\pi) + 5dh)$. Furthermore, for $n \geq 1$ and $j \geq 2$, by $\binom{m}{n} \leq (\frac{em}{n})^{n}$, if $j \geq n$, we have

$$\mathcal{E}_{n}^{j} \leq \sum_{a=2}^{n} 0.02^{j-1}2^{n-a}\binom{n-a+j-2}{j-2}\mathcal{E}_{a}^{1} + \sum_{b=2}^{j}\binom{n+j-b}{j-b}0.02^{j-b}2^{n}\mathcal{E}_{0}^{b}$$

$$\leq (2h^{2}\mathsf{FI}(\mu^{0}\|\pi) + 5dh)\left(\sum_{a=2}^{n} 0.02^{j-1}2^{n-a}\binom{n-a+j-2}{j-2}2^{a+1} + \sum_{b=2}^{j}\binom{n+j-b}{j-b}0.02^{j-b}2^{n}0.01^{b-1}\right)$$

$$\leq (2h^{2}\mathsf{FI}(\mu^{0}\|\pi) + 5dh)0.02^{j-1}2^{n+1}\left(\sum_{a=2}^{n}\binom{n-a+j-2}{j-2} + \sum_{b=2}^{j}\binom{n+j-b}{j-b}\right)$$

$$\leq (2h^{2}\mathsf{FI}(\mu^{0}\|\pi) + 5dh)0.02^{j-1}2^{n+1}\left(\sum_{a=2}^{n}e^{j-2}\left(\frac{n-a+j-2}{j-2}\right)^{j-2} + \sum_{i=0}^{j-2}e^{j-2}\left(\frac{n+i}{i}\right)^{i}\right)$$

$$\leq (2h^{2}\mathsf{FI}(\mu^{0}\|\pi) + 5dh)0.02^{j-1}2^{n+1}\left(\sum_{a=2}^{n}e^{j-2}2^{j-2} + \sum_{i=0}^{j-2}e^{j-2}e^{n}\right)$$

$$\leq (2h^{2}\mathsf{FI}(\mu^{0}\|\pi) + 5dh)0.02^{j-1}2^{n+1}e^{2j}(2j)$$

$$\leq 200(2h^{2}\mathsf{FI}(\mu^{0}\|\pi) + 5dh)0.5^{j}2^{n},$$

where the last equation holds since $0.02^{j-1}e^{2j}(2j) \leq 100 \cdot 0.5^{j}$. $\square$

## A.4 Overall bound

Combine Equation (2) and Lemma A.5, we have

$$\frac{1}{Nh}\int_0^{Nh}\mathsf{FI}(\mu_t\|\pi)\mathrm{d}t \le \frac{2\mathsf{KL}_{0,0}}{Nh} + \frac{2Ld}{M} + 600L^2(2h^2\mathsf{FI}(\mu^0\|\pi) + 5dh)0.5^{J-N}.$$

By the convexity of the Fisher information, the averaged distribution $\bar{\mu} := (Nh)^{-1}\int_0^{Nh}\mu_t\mathrm{d}t$ also satisfies

$$\mathsf{FI}(\bar{\mu}\|\pi)\mathrm{d}t \le \frac{2\mathsf{KL}_{0,0}}{Nh} + \frac{2Ld}{M} + 600L^2(2h^2\mathsf{FI}(\mu^0\|\pi) + 5dh)0.5^{J-N}.$$

We also observe the output is sampled from $\bar{\mu}$. By Lemma A.2, and $Lh = 0.1$ we have

$$
\begin{aligned}
\mathsf{FI}(\bar{\mu}\|\pi)\mathrm{d}t &\le \frac{2L\mathsf{KL}_{0,0}}{N} + \frac{2Ld}{M} + 600L^2\left(2h^2\left(\frac{2\mathsf{KL}(\mu_0\|\pi)}{Nh} + 8L^2dh\right) + 5dh\right)0.5^{J-N} \\
&\le \frac{2L\mathsf{KL}_{0,0}}{N} + \frac{2Ld}{M} + \left(\frac{24L\mathsf{KL}(\mu_0\|\pi)}{N} + 320Ld\right)0.5^{J-N}.
\end{aligned}
$$

# B  Limitations

As an initial exploration of the adaptive complexity of minimizing relative Fisher information, our work still leaves a significant gap between the upper and lower bounds, particularly in the small-accuracy regime.

# C  Social Impacts

We present several theoretical results for minimizing relative Fisher information. While we do not see any immediate societal impact, there may be potential indirect consequences of our work that are not apparent at this time.

