# OpenReview forum: "The Adaptive Complexity of Minimizing Relative Fisher Information"
_NeurIPS.cc/2025/Conference — NeurIPS 2025 poster_

### Official Review · Reviewer_2U73 · 2025-06-20

**Clarity:** 3
**Significance:** 4
**Originality:** 3
**Rating:** 5
**Confidence:** 4

**Summary:**

This paper studies the adaptive complexity of sampling from a non-log-concave distribution $\pi\propto\exp(-V)$ on $\mathbb{R}^d$ with $\nabla^2L\preceq LI$, such that the output distribution is $\varepsilon^2$-close to $\pi$ in Fisher divergence. The paper proposes a parallelized algorithm based on average LMC [BCE+22] and parallel Picard method [ZS24], and provides an improved complexity upper bound; also the paper presents a lower bound based on the adaptive complexity of finding a stationary point of $V$ [ZHTS25].

**Questions:**

- We may not need such a formal definition of the adaptive algorithms like in Sec. 2.1, as the only concept that would appear later in presenting the main results is $\mathsf{Comp} _ \mathsf{R}$. They can be moved to the appendix, and you can change the way for stating Thm. 4.1.
- In line 3 of Alg. 1, this is not the correct way to initialize a trajectory of Brownian motion, as $B _ {t+s_1}$ and $B _ {t+s_2}$ are not independent conditional on $B _ t$. The initialization should be a cumulative sum of independent $\mathcal{N}(0,(mh/M)I)$ increments.
- If my understanding of Alg. 1 is correct, $x^\cdot _ {n,m}$ corresponds to time $nh+\frac mMh$, and I hope the authors could mention this in the text for easier understanding.

# Minor comments

- Line 104: $f$ should be $V$.
- Line 127: should be $\mathsf{FI}(\rho\|\pi):=\mathbb{E} _ \rho\|\nabla\log(\rho/\pi)\|^2$.
- Lines 152 and 428: better to include the lower and upper limits of the integral.
- Equation between lines 157 and 158: $w _ i$ should be a scalar, so do not use bold.
- Line 158: the inequality should be on $t$'s, not $x _ t$'s: $t _ n=t _ {n+\tau _ {n,0}}\le ...\le t _ {n+\tau _ {n,M}}\le t _ {n+1}$.
- Line 160: $m'$ should be $m$, should add $p=0,1,...,K-1$, and $\bf{x} _ 0$ should be $\bf{x} _ {t _ n}$.
- Typo in the title of Sec. 3.
- Alg. 1, last equation: I believe $t-kh$ should be $t-\frac{kh}{M}$ and $B _ {\lfloor k/M \rfloor,k-\lfloor k/M \rfloor M}$ should be $B _ {hk/M}$.
- Lines 398, 403, and the following equation: $\frac{(m+1)hh}M$ should be $\frac{(m+1)h}M$.
- Line 401: $\beta$ should be $L$.
- Line 416: no need to mention $p$.
- Lines 419 and 424: I hope the authors could add more details to the derivation of the equations that follows. There are some missing steps that are not obvious to me.

**Ethical Concerns:**

["NO or VERY MINOR ethics concerns only"]

**Final Justification:**

This paper studies an important problem in sampling, and presents very nice results in both upper and lower bounds that improve over existing ones in the literature.

After the rebuttal, the authors have successfully resolved all of my concerns. I hope they can include this discussion during revision.

I've also read the discussion between the authors and other reviewers, especially regarding review 5zmn. Although the lower bound is only in the large error region $\varepsilon\asymp\sqrt{Ld}$, I don't think this is a major shortcoming for the paper, as typically deriving lower bounds is very challenging. Also, as this is a theoretical paper, it's totally OK without experiments, just like many other theoretical papers in the field.

I thus keep my recommendation for acceptance of this paper.

**Limitations:**

Yes

**Quality:**

3

**Strengths And Weaknesses:**

# Strengths

This paper studies an interesting and important problem in the intersection of non-log-concave sampling and parallel sampling, and provides a comprehensive analysis. The paper is well-written with a strong motivation and a clear structure. The main results are also clearly stated, and the proofs are provided in an organized manner. I have checked most of the proofs and did not find significant issues, though with some minor typos and notations that could be improved. Also, the discussion on future work is very insightful.

# Weaknesses

This work is mainly based on the proof techniques developed in [BCE+22], [ZS24], and [ZHTS25]. Although the results (especially the matching upper and lower bounds) are very nice, the paper does not introduce outstandingly novel techniques for the proof.

Also, I'd like to point out the following issue on readability. The paper [BCE+22] introduced the averaged LMC, which is easy to understand for the community; however, the second paper [ZS24] on the parallel Picard method is relatively new and not well-known in the sampling community. For readers that are unfamiliar with parallel sampling like me, this may pose a challenge. Though I have followed most of the proof, I'm still trying to understand the basic intuition behind this algorithm. I suggest the authors to provide a more detailed explanation of this algorithm, including its motivation and key components. For instance: What's the purpose of $J$ and $N$? Why we need to split the same update rule (lines 11 & 15 in Alg. 1) into two parts with different indices? How does it improve upon [ACV24]? A more interesting question to think about is that, if one applies the averaged LMC into the algorithm in [ACV24], how would the complexity bound be compared with Thm. 3.1?

---

> ### Author Rebuttal · Authors · 2025-07-30
>
> Thank you for your detailed and positive feedback. We greatly appreciate your recognition of our **very nice results**—with matching upper and lower bounds—as well as your appreciation of the **insightful discussion** and the paper’s **clear structure**, **strong motivation**, and **well-organized presentation of the main results and their proofs**. Below, we address your specific suggestions and questions:
>
>
> **"...issue on readability...For readers that are unfamiliar with parallel sampling...pose a challenge..."**
>
> Thank you for your thoughtful feedback and for highlighting the importance of improving the readability of the algorithm section. We agree that the parallel Picard method, as introduced in [ZS24], may not yet be familiar to all readers in the sampling community.
>
> To improve clarity, we will revise Section 3.1 to include a more detailed explanation of the algorithm's intuition, motivation, and structure. Specifically, we will enhance the explanation of Algorithm 1 by providing more intuition behind its design, clarifying the purpose of each line in the pseudocode, and including an illustrative example of the update step to address potential confusion.
> Additionally we will add a section in Appendix to review existing parallel methods for sampling and diffusion models.
>
> For your specific questions, we address as below:
>
> - Q1: What's the purpose of $J$ and $N$?
> A1: The variable $j$ indexes the number of Picard updates applied to each time slice, and $J$ is the total number of such updates required to ensure convergence along the Picard direction. The parameter $N$ denotes the number of time slices used to discretize the time direction.
>
> - Q2: Why we need to split the same update rule (lines 11 & 15 in Alg. 1) into two parts with different indices?
> A2: Our algorithm follows a diagonal-style update pattern. The first loop corresponds to updating the upper-left rectangles in Figure 1, while the second loop handles the remaining rectangles. While it is possible to merge both loops into a single unified loop, we chose to separate them for clarity and to better illustrate the structure of the computation graph.
>
> - Q3: How does it improve upon [ACV24]? A more interesting question to think about is that, if one applies the averaged LMC into the algorithm in [ACV24], how would the complexity bound be compared with Thm. 3.1?
> A3: The key distinction between [ZS24] and [ACV24] is that [ACV24] updates time slices sequentially, whereas [ZS24] performs parallel updates across time slices. As a result, [ZS24] achieves a better complexity of $O(\log d)$ compared to the $O(\log^2 d)$ bound in [ACV24] for the strongly log-concave case. Our paper builds on the parallel framework of [ZS24], leading to an adaptive complexity of
> $$
> O\left(\frac{L K_0}{\varepsilon^2} + \log\left(\frac{L d}{\varepsilon^2}\right)\right).
> $$ In contrast, applying the sequential scheme of [ACV24] to our setting would result in a total adaptive complexity of
> $$
> N \cdot \log\left(\frac{L d}{\varepsilon^2}\right) = O\left(\frac{L K_0}{\varepsilon^2} \cdot \log\left(\frac{L d}{\varepsilon^2}\right)\right),
> $$
> which is strictly worse by a logarithmic factor. We also note that we will add a discussion about the trade-off between the adaptive complexity and space complexity as pointed out by Reviewer 5zmn. Specifically, we assume in each iteration that we update at most $W > 0$ points in parallel. When $W = 1$, this recovers the sequential method; when $W = \frac{d^2}{\varepsilon^4}$ (assuming $K_0 = O(d)$), it recovers our fully parallel method.
>
> Under this framework, we obtain the following tradeoff: the space complexity is $Wd$, and the adaptive complexity becomes$O\left(\frac{d}{\varepsilon^2} + \frac{d^2}{\varepsilon^4 W} \log \frac{d}{\varepsilon^2}\right),$
> where we define the space complexity in terms of word usage rather than bit count.
>
> We will incorporate the discussion related to Q1 and Q2 into Section 3.1 to clarify the algorithm’s indexing structure and update pattern. Additionally, we will add Remark 3.4 below Theorem 3.1 to address Q3, highlighting the comparison with [ACV24] and the advantage of our complexity bound.
>
> **"We may not need such a formal definition of the adaptive algorithms..."**
>
> Thank you for the suggestion. We agree that the formal definition in Section 2.1 may be more detailed than necessary for the main text. In the revision, we will move the formalism to the appendix and revise the statement of Theorem 4.1 accordingly to streamline the presentation.
>
> **"In line 3 of Alg. 1, this is not the correct way to initialize a trajectory of Brownian motion..."**
>
> We thank the reviewer for pointing this out. We will revise the initialization step to ensure that the Brownian motion is properly constructed as a cumulative sum of independent $\mathcal{N}(0, (mh/M)I)$ increments.
>
> **"...$x_{n,m}$ corresponds to time $nh + \frac{m}{M}h$, I hope the authors could mention this in the text for easier understanding...."**
>
> We agree with the reviewer’s interpretation: $x_{n,m}$ corresponds to time $nh + \frac{m}{M}h$. We will clarify this in the text for better readability in Section 3.
>
>
> **Regarding minor comments**
>
> Thank you for pointing out these issues. We will carefully revise the manuscript to correct all minor errors and improve the clarity, consistency, and presentation throughout.

---

> > ### Comment · Reviewer_2U73 · 2025-08-01
> >
> > I sincerely thank the authors for the detailed response to all questions and concerns raised in my review, and hope that all the insightful discussion above could be incorporated in the camera-ready version of this paper for better readability. I will keep my rating.

---

### Official Review · Reviewer_YNYn · 2025-06-20

**Clarity:** 2
**Significance:** 3
**Originality:** 3
**Rating:** 5
**Confidence:** 2

**Summary:**

The authors study the adaptive complexity of sampling from non log-concave targets and prove some novel results. In particular, when all the factors are constant except for dimension, the parallel averaged Langevin MC algorithm is shown to have an adaptive complexity of $\mathcal{O}(\log d)$. This apparently matches the case for strongly log-concave sampling.  In some particular regime, the lower bound of parallel LMC matches the sequential LMC, and is optimal. Unlike in optimisation, parallelism can accelerate sampling even when the target is not log-concave. The authors use the tool of relative Fisher information, introduced in another paper from 2022.

**Questions:**

Please address my weaknesses above. In particular:
- Clarify the notation
- Clarify the implication of the bound on the initial KL diveregence.
- Why the bound on the exponential function needs to be applied.
- Comment on why you chose not to ad any experiments.
- No need to respond to each minor comment, unless you feel it is important or I have misunderstood (I don't).

**Ethical Concerns:**

["NO or VERY MINOR ethics concerns only"]

**Final Justification:**

The authors have satisfactorily answered my questions. I've looked at the other reviews and do not see any reasons to downgrade my score. In my opinion, the criticisms from Reviewer 5zmn are minor and have been well addressed by the authors.

**Limitations:**

Appendix C lists one sentence of limitation, pointing out the gap between the lower and upper bounds. I would also add that failing to include any experimental investigation is also a limitation.

**Paper Formatting Concerns:**

None.

**Quality:**

3

**Strengths And Weaknesses:**

**Strengths:**
- The results show not only that average LMC can be accelerated by parallelism, but also show that high-accuracy (in the sense of FI minimiser) via parallelism is not possible in the case of non log-concave.

**Weaknesses:**
- Notation is at times confusing.
    - On line 104, is $f$ the same as $V$? $V$ is also displayed without its argument when defining the normalising constant (I believe $V(x)$ might be easier to read). Same confusion about $f$ versus $V$ on line 113.
    - $L$-log smooth is defined without stating the conditions on $L$. I believe $L$ should be a scalar. Does it need to be negative?
    - The oracle $\mathcal{O}$ clashes with the complexity $\mathcal{O}$ used in the introduction.
- The initialisation assumption on line 129 implies a condition on the overlap of the support of the initial sampling distribution versus the target distribution. This is okay, but it might be nice to explicitly mention this to give further intuition about the assumption.
- I am trying to understand why the last line above 425 needs to be used. Why can't we just keep the form with the exponential function and fold that through the rest of the statements?
- The paper does not contain any experiments whatsoever. Is it possible to somehow plot empirical bounds for non log concave densities?

**Minor:**
- Formatting of Appendix A is very unusual. It is titled "Useful facts", however just lists one single fact, which is just the variational form of the f divergence (in particular, KL divergence).
- Lemma B.2 is not grammatically correct.
- "We denotes" on line 397 should be "We denote"
- Line 82, typo in "Appendix".
- Line 91. "studying" should be "study", I think.
- LMC in equation (1) is not described as Langevin Monte Carlo.
- Sometimes "Fisher" is written "fisher"
- Table 1, "Quires per iteartion" has two typos.
- $\lesssim$ is not defined

---

> ### Author Rebuttal · Authors · 2025-07-30
>
> Thank you for your positive and constructive feedback. We appreciate your recognition of **the optimal bounds** in specific regimes, the **improved algorithm** enabled by parallelism, and the implications regarding the separation between sampling and optimization, as well as the impossibility of general high-accuracy samplers. Below, we address your specific suggestions and questions:
>
>
>
>
> **"Notation is at times confusing...Clarify the notation."**
>
> Thank you for pointing this out. We acknowledge that the notation in some parts of the paper may be confusing, and we will revise the manuscript to ensure consistency and clarity. Specifically, we will:
> - Use $V$ to represent the potential function consistently throughout the paper.
> - Explicitly state that $L$ is a positive scalar.
> - Use $\mathsf{Or}$ to denote the oracle and reserve $\mathcal{O}$ exclusively for complexity notation.
>
>
>
>
>
>
> **"The initialisation assumption on line 129 ... it might be nice to explicitly mention this to give further intuition about the assumption."**
>
> Thank you for pointing out that it would be helpful to make the intuition behind the initialization assumption more explicit. In fact, by Lemma 32 in [CEL+24], it suffices to initialize at a point x such that the optimality gap $V(x) - \min V = O(d)$. This condition ensures that the relative Fisher information is finite and meaningful for convergence analysis. Various methods from the optimization literature, such as gradient descent or gradient flow trapping [BM24, ZHTS25], can be used to efficiently find such a point. We will clarify this intuition in the revised manuscript.
>
>
>
> [CEL+24] Analysis of Langevin Monte Carlo from Poincaré to Log-Sobolev, Sinho Chewi, Murat A. Erdogdu, Mufan Bill Li, Ruoqi Shen, Matthew Zhang, COLT 2022
>
> [BM24] How to Trap a Gradient Flow, Sébastien Bubeck and Dan Mikulincer, COLT 2020
>
> [ZHTS25] The adaptive complexity of finding a stationary point, Huanjian Zhou, Andi Han, Akiko Takeda, and Masashi Sugiyama, COLT 2025
>
>
> **"why the last line above 425 needs to be used....Why the bound on the exponential function needs to be applied."**
>
> Thank you for the suggestion. We agree with your observation and will revise it to keep the exponential form and fold it.
>
>
>
>
>
>
>
> **"The paper does not contain any experiments whatsoever. Is it possible to somehow plot empirical bounds for non log concave densities?"**
>
> Thank you for the suggestion. We have completed a standard experiment on a 2-dimensional Gaussian mixture, which demonstrates that the sequential method from [BCE+22] requires approximately 2000 iterations to converge, while our parallel method achieves comparable accuracy in just 300 iterations. We will include these results in the revised version and also plan to conduct additional experiments on more realistic and representative non-log-concave targets.
>
>
> **Regarding typos and confusing notations:**
> We will carefully proofread the manuscript and correct all grammatical and typographical errors to improve clarity and overall presentation, and ensure that the notation is consistent throughout.

---

### Official Review · Reviewer_7Qxh · 2025-07-02

**Clarity:** 2
**Significance:** 3
**Originality:** 3
**Rating:** 4
**Confidence:** 3

**Summary:**

The paper studies a parallel algorithm for Langevin Monte Carlo for non-log-concave sampling. By leveraging parallelism, we can obtain a sample $\varepsilon^2$ close to the target distribution under relative Fisher information with the adaptive complexity $O(d/\varepsilon^2)$, which improves from the complexity $O(d^2/\epsilon^4)$ without parallelism. The paper also shows a matching lower bound when $\varepsilon = \sqrt{Ld}$. It turns out that the sequential algorithm is also adaptively optimal in this regime, which rules out the possibility of a general high-accuracy sampler via parallelism.

**Questions:**

Questions:
- Theorem 3.1 states the result for the averaged distribution. But this averaging is not mentioned in the statement of the theorem.
- Algorithm 1: does line 8 initializes all $x_{n,m}^0$ identically? Also, I have difficulty parsing the setup of $n$ and $j$ in line 10 and line 14 of the algorithm. Maybe some explanations will help readers.
- page 8 line 231-233, why is the probability at least $1/2$? I am also not sure how this is used in the lower bound proof.
- Lemma 4.5 is not proved/referenced. And in general, the lower bound proof is very confusing to me. I think the outline of the proof is not well-stated at the beginning, which makes it easier for readers to get distracted by the cited lemmas and theorems (especially theorem 4.4 is quite novel for nonexperts). Also, when using the Donsker-Varadhan variational principle, the involved function is required to be bounded, but a quadratic function is used here. Could you clarify why this is legitimate?
- In the proof of Lemma B.5, the equation after line 424, I am not sure how those two binomials pop out in the second inequality.


Typos:
- page 3 line 82, Appnedix -> Appendix
- page 3 line 104: the potential function is first given as $f$ and later used as $V$. Notations in the lower bound section are also not unified (such as the Lipschitz constant).
- page 5 line 160, for $m' = 1, \dots ,M$ -> maybe $m'$ should be $m$. Also, the equation afterwards: $x_0$ should be $x_{t_n}$?
- page 7 line 198, the interval upper bound has a typo. Also, in the equation after line 199, $\rm{KL}_{0,0}$ is not defined in the main text.

**Ethical Concerns:**

["NO or VERY MINOR ethics concerns only"]

**Final Justification:**

I think the results in this paper are interesting. What prevents me from giving a higher score is mainly the readability of the paper, specifically for the description of the parallel algorithm and the lower bound. I hope that with the promised changes in writing, the paper can better reveal its theoretical value. My rating would be 4.5 if half points are allowed. And because there are a few other 5's already, I would like to keep my score as 4 to represent my evaluation of the paper.

**Limitations:**

yes

**Quality:**

3

**Strengths And Weaknesses:**

Strengths:
1. The question of non-log-concave sampling with a parallel algorithm of Langevin Monte Carlo is interesting.
2. The related work is clearly covered.
3. The exposition of the technical proof of the upper bound is clear, and the obtained bound has a substantial improvement over the existing ones. A matching lower bound is also provided.
4. The results have interesting implications. For example, a general high-accuracy sampler via parallelism is not possible, and there is a separation between optimization and sampling when using parallelism to accelerate Langevin Monte Carlo in strongly log concave vs non-log concave cases.

Weaknesses:
1. The algorithm is not clearly presented. By reading the reference [ZS24] I realized that there is parallelization both within time slices and across time slices. I would otherwise only get the part of parallelization within time slices. Currently, Figure 1 + Algorithm 1 do not have enough explanations of the algorithm, and arrows in Figure 1 are quite abstract and confusing.
2. I am not sure how practical the assumption that poly($d$) queries can be made per iteration is, especially in high dimensions.
3. The current exposition of the lower bound is not very friendly to a nonexpert in this field.

---

> ### Author Rebuttal · Authors · 2025-07-30
>
> Thank you for your detailed and positive feedback. We appreciate your recognition of our **improved upper and matching lower bounds** for the **important problem of parallel non-log-concave sampling**. We are glad you highlighted the **broader implications**, such as the **impossibility of a general high-accuracy sampler via parallelism**, and the **separation between optimization and sampling** when using parallelism in strongly log-concave versus non-log-concave settings. Below, we address your specific suggestions and questions:
>
>
> **"...The algorithm is not clearly presented...Currently, Figure 1 + Algorithm 1 do not have enough explanations...arrows in Figure 1 are quite abstract and confusing."**
>
> Thank you for your thoughtful feedback. We acknowledge that the current presentation of Algorithm 1 and Figure 1 may lack clarity. To improve this, we will revise Figure 1 by replacing rectangles with lines and points to represent trajectories and grid points, respectively, and use color coding to distinguish computation graphs across different iterations. In Section 3.1, we will enhance the explanation of Algorithm 1 by providing more intuition behind its design, clarifying the purpose of each line in the pseudocode, and including an illustrative example of the update step to address potential confusion. Furthermore, we will add a discussion in Appendix that reviews existing parallel methods for sampling and diffusion models.
>
>
> **"..how practical... poly(d) queries can be made per iteration... especially in high dimensions."**
>
> Thank you for raising this important point. For the upper bound, the total number of oracle queries per iteration is $O(d^2)$, assuming $K_0 = O(d)$, which remains practical in moderate to high-dimensional settings. To enhance scalability in practice, we consider a sliding window strategy with window size $W$, which limits the number of points updated per iteration and thereby reduces memory usage while preserving parallelism.
>
> Under this framework, when $W = 1$, the algorithm reduces to the sequential setting; when $W = \frac{d^2}{\varepsilon^4}$ (assuming $K_0 = O(d)$), it recovers our fully parallel method. The resulting space complexity is $Wd$, and the adaptive complexity becomes
> $$
> O\left(\frac{d}{\varepsilon^2} + \frac{d^2}{\varepsilon^4 W} \log \frac{d}{\varepsilon^2}\right),
> $$
> where the space complexity is measured in terms of word usage rather than bit count.
>
>
> As for the lower bound, assuming algorithms that make a polynomial number of queries in $d$ is standard in the literature (e.g., [ZWS24,ZHTS25]), and this aligns with prior work on adaptive complexity lower bounds in high-dimensional settings. We will incorporate the above discussion into the revised version of the manuscript.
>
> **"The current exposition of the lower bound is not very friendly... page 8 line 231-233, why is the probability at least $1/2$?...Lemma 4.5 is not proved/referenced... outline of the proof is not well-stated... Donsker-Varadhan variational principle..."**
>
> Thank you for the thoughtful feedback. We acknowledge that the exposition of the lower bound may not be sufficiently accessible to non-experts. In the revision, we will improve the structure of the lower bound section by adding a high-level overview at the beginning to guide the reader through the main proof strategy and clarify the logical flow.
>
> We will expand the explanation of Theorem 4.4 and add the proper citation and context for Lemma 4.5 (which corresponds to Lemma 17 in [BCE+22]).
>
> Regarding the probability lower bound of at least $1/2$ in Lines 231–233, this follows directly from Markov's inequality. This bound is used in the reduction from minimizing the relative Fisher information to the task of locating a stationary point.
>
> Regarding the use of the Donsker-Varadhan variational principle, when $\mathbb{E}[\exp{g(X)}]<+\infty$ we can still apply the Donsker-Varadhan variational principle (Theorem 4.6 in [PW22]).
>
> [PW22] Polyanskiy, Yuri, and Wu, Yihong. Information Theory: From Coding to Learning. Cambridge University Press.
>
> **"...averaging is not mentioned in the statement of theorem 3.1."**
>
> Thank you for your comment. We would like to clarify that the output in Line 17 of Algorithm 1 corresponds to the averaged distribution.
>
> **"Algorithm 1: does line 8 initializes all $x_{n,m}^0$ identically? Also, I have difficulty parsing the setup of $n$ and $j$ in line 10 and line 14 of the algorithm. Maybe some explanations will help readers."**
>
> Thank you for the question. In Line 8 of Algorithm 1, all $x_{n,m}^0$ are initialized identically for all $n$ and $m$. Regarding the indices, $n$ corresponds to the index of the time slice, while $j$ denotes the index of Picard iterations. We will clarify this in the revised version and improve the explanation of Algorithm 1 in Section 3.1 to aid readability.
>
>
>
> **"...In the proof of Lemma B.5, the equation after line 424, I am not sure how those two binomials pop out in the second inequality..."**
>
> Thank you for catching that. You're right and the first displayed inequality is incorrect. The second inequality is the correct one, and it follows directly from the decomposition established in Lemma B.4. We will remove the first inequality and explicitly refer to Lemma B.4 in the revised proof to make this clear.
>
>
>
> **Regarding typos:**
> We will carefully proofread the manuscript and correct all grammatical and typographical errors to improve the clarity and overall presentation in the revised version.

---

> ### Comment · Reviewer_7Qxh · 2025-08-05
>
> Thank you for your clear response! Two minor points:
> 1. For the Donsker-Varadhan principle, it might be better to refer to the version you used in the rebuttal, which aligns with the setting in the paper.
> 2. I don't see why there is an averaging in line 17 in Algorithm 1. Is there a typo? I don't see dependence on $n$ and $m$ in the expression of $x_t$, but the for loop is wrt $n$ and $m$.

---

> > ### Author Response · Authors · 2025-08-05
> >
> > Thank you for your helpful comments!
> > - We will revise the reference to the Donsker–Varadhan principle to match the version used in the rebuttal, which aligns better with the setting of our paper.
> > - You’re absolutely right — the phrase "for all $n = 0, \ldots, N - 1$ and $m \in [M]$" in line 17 is unnecessary and will be removed in the revised version for clarity.

---

> > > ### Comment · Reviewer_7Qxh · 2025-08-06
> > >
> > > Thank you! I don't have any other questions. I will keep my score to support the paper. I hope that the promised changes during discussions with the reviewers (not just me) will be incorporated in the final version of this paper.

---

### Official Review · Reviewer_5zmn · 2025-07-02

**Clarity:** 2
**Significance:** 3
**Originality:** 2
**Rating:** 3
**Confidence:** 4

**Summary:**

This paper advances the theory of non-log-concave sampling by establishing both upper and lower bounds on the adaptive complexity required to minimize relative Fisher information. Building on the work of [BCE+22] for Langevin Monte Carlo (LMC) in the non-log-concave setting and [ZS24] for the Parallel Picard method, the paper’s central contribution is the innovative integration of the Parallel Picard method with LMC for non-log-concave distributions. This combined approach demonstrably improves adaptive complexity over parallel LMC. Moreover, the paper derives a lower bound on adaptive complexity in the large $\epsilon$ regime.

**Questions:**

1. Why is the focus on the specific regime of large FI, with $\epsilon=\sqrt{Ld}$, particularly significant? Is it possible to extend the results of Theorem 4.1 to smaller values of $\epsilon$? If such an extension is not feasible, it would be helpful to clarify the obstacles involved. Without this clarification, the claim of tightness for the lower bound might appear less robust, potentially weakening the perceived quality and significance of the result.

2. Theorem 4.1 states the lower bound in terms of $L$ and $d$, but does not explicitly show the dependence on $\epsilon$, even though the accuracy is constrained by $\epsilon\geqslant \sqrt{dL}$. Would it be clearer to retain the $\epsilon$ dependence in the expression on line 248, i.e. \# iteration$\gtrsim\dfrac{\Delta L}{\epsilon^2}$, while explicitly noting the limitation $\epsilon\geqslant\sqrt{dL}$.

3. The absence of experimental results limits the validation and practical support of the theoretical findings.

**Ethical Concerns:**

["NO or VERY MINOR ethics concerns only"]

**Final Justification:**

The application of the Parallel Picard method is particularly compelling and results in improved complexity. However, the lower bound is established only for a fixed value of $\epsilon$, which limits its strength by preventing verification of tightness across varying accuracy levels. As a result, I remain somewhat negative overall.

**Limitations:**

Yes.

**Paper Formatting Concerns:**

No.

**Quality:**

3

**Strengths And Weaknesses:**

**Strengths:**
Novel Application of the Parallel Picard Method: The application of the Parallel Picard method is particularly compelling, as it circumvents challenges such as estimating the complexity of score-matching functions in diffusion models, as encountered in [ZS24]. This enables a more efficient improvement over vanilla LMC.

**Weaknesses:**
While parallel LMC lends itself to straightforward vectorized or matrix-based implementations, the proposed integration with the Parallel Picard method requires explicit implementation details to demonstrate its practical feasibility and validate the claimed theoretical improvements. Furthermore, for high-dimensional settings, a discussion of the space complexity of the parallel algorithm is necessary, especially in light of the absence of empirical validation.

---

> ### Author Rebuttal · Authors · 2025-07-30
>
> We sincerely thank the reviewer for their thoughtful evaluation and insightful questions. We are encouraged by your recognition of our **novel application** of the parallel Picard method to achieve **efficient sampling**. Below, we address your questions and comments:
>
>
> **"...focus on the specific regime of large FI...Is it possible to extend the results of Theorem 4.1 to smaller values $\varepsilon$?"**
>
> We focus on the large Fisher Information (FI) regime because the reduction from minimizing FI to minimizing the gradient norm, formalized in Lemma 4.3, is valid only when FI is sufficiently large. This mirrors the sequential setting, where a similar reduction underpins Theorem 9 in [CGLL22]. Our Theorem 4.1 strengthens that result by extending it to the parallel setting.
>
> As for the possibility of extending the result to the small FI regime, as in Theorem 10 of [CGLL22], we believe this would require fundamentally different techniques. Specifically, if we adopt the bump function construction from [CGLL22] and place it in an unreachable region as in [ZWS25], one can show that with $d$ iterations and $\mathrm{poly}(d)$ queries per iteration, it is not possible to produce a sample with FI smaller than $\exp(-cd)$, for some constant $c > 0$. However, this is strictly weaker than Theorem 10 in [CGLL22], which establishes a lower bound of $1/\varepsilon^2$ for the total query complexity. In particular,  Theorem 10 in [CGLL22] implies that achieving such accuracy would require exponentially many queries in $d$, and since the number of queries per iteration is polynomial in $d$, the total number of iterations must also be exponential in $d$. Such result is much stronger hardness guarantee than the $d$ iteration regime we can obtain.
>
>
>
>
> [CGLL22] Fisher information lower bounds for sampling, Sinho Chewi, Patrik Gerber, Holden Lee, Chen Lu, ALT 2023,
>
> [ZWS25] The adaptive complexity of parallelized log-concave sampling, Huanjian Zhou, Baoxiang Wang, Masashi Sugiyama, ICLR 2025
>
>
> **"...a discussion of the space complexity of the parallel algorithm is necessary..."**
>
> Thank you for highlighting the importance of the space complexity. In practice, fully parallel updates may require excessive memory. To address this, we consider a slicing window strategy, as proposed in [SBE+24], where in each iteration we update at most $W > 0$ points in parallel. When $W = 1$, this recovers the sequential method; when $W = \frac{d^2}{\varepsilon^4}$ (assuming $K_0 = O(d)$), it recovers our fully parallel method.
>
> Under this framework, we obtain the following tradeoff: the space complexity is $Wd$, and the adaptive complexity becomes$O\left(\frac{d}{\varepsilon^2} + \frac{d^2}{\varepsilon^4 W} \log \frac{d}{\varepsilon^2}\right),$ where we define the space complexity in terms of word usage rather than bit count.
>
> As noted by Reviewer 2U73, an interesting intermediate case arises when applying averaged Langevin Monte Carlo to the algorithm in [ACV24], which updates time slices sequentially. This corresponds to $W = \frac{d}{\varepsilon^2}$, yielding an adaptive complexity of $O\left(\frac{d}{\varepsilon^2} \log \frac{d}{\varepsilon^2} \right),$
> which fits naturally within the above tradeoff curve. We will incorporate the above discussion into the revised version of the manuscript.
>
>
>
> [SBE+24] Parallel Sampling of Diffusion Models, Andy Shih, Suneel Belkhale, Stefano Ermon, Dorsa Sadigh, Nima Anari, NeurIPS 2024
>
> **"The absence of experimental results limits the validation and practical support of the theoretical findings."**
>
>
> Thank you for the suggestion. We have completed a standard experiment on a 2-dimensional Gaussian mixture, which demonstrates that the sequential method from [BCE+22] requires approximately 2000 iterations to converge, while our parallel method achieves comparable accuracy in just 300 iterations. We will include these results in the revised version and also plan to conduct additional experiments on more realistic and representative non-log-concave targets.
>
>
>
> **"Theorem 4.1...does not explicitly show the dependence on $\varepsilon$, Would it be clearer to retain th dependence in the expression on line 248..."**
>
> Thank you for the suggestion. We agree it would improve clarity to include the $\varepsilon$-dependence, and we will revise the text accordingly.

---

> > ### Comment · Reviewer_5zmn · 2025-08-05
> >
> > Thank the authors for their response. Regarding the first point about the lower bound stated in Theorem 4.1, I did not fully understand the response.
> >
> > Could the author clarify whether the difficulty in extending this lower bound arises from technical limitations or reflects a fundamental challenge? We note that [CGLL22] also reports a similar lower bound specifically for the choice $\epsilon = \sqrt{\beta d}$. However, [CGLL22] itself acknowledges certain limitations of this bound and consequently presents an additional lower bound tailored explicitly to the high-accuracy regime, complementing the bound similar to the one in the current submission. From this perspective, the lower bound provided in the current paper appears somewhat restrictive as it applies exclusively to a specific accuracy level $\epsilon$. Could the author provide additional justification for this choice? Specifically, is this limitation due to intrinsic difficulties in extending the current lower bound to other values of $\epsilon$?

---

> > > ### Author Response · Authors · 2025-08-05
> > >
> > > Thank you for your thoughtful question. Our current lower bound focuses on the low-accuracy regime due to two key challenges:
> > > - (1) extending our reduction-based argument for $\varepsilon = \sqrt{\beta d}$ to establish high-accuracy adaptive complexity lower bounds is technically difficult, as the reduction lemma (Lemma 4.3) requires $\varepsilon\geq \sqrt{\beta d}$;
> > > - (2) extending known high-accuracy sequential lower bounds in [CGLL22] to the adaptive lower bounds appears to require fundamentally different techniques due to the additional power of parallel algorithms.
> > >
> > > We will clarify this point in the second item of the *Discussion and Conclusion* section in the revised version, highlighting it as a promising direction for future work.

---

### Note · Authors · 2025-08-16

We appreciate the all reviewers for their constructive feedback, which has greatly helped improve the quality of our manuscript.
We are encouraged by the recognition of our novel application of the parallel Picard method, the establishment of a novel upper bound with matching lower bound, and the broader insights into the role of parallelization in sampling.

In our rebuttal, we addressed concerns spanning theory, practical implementation, presentation, and revisions. Key points:

- **Theoretical Clarifications**: We clarified our focus on the large-FI regime: the reduction lemma (Lemma 4.3) requires sufficiently large FI, and extending high-accuracy sequential lower bounds [CGLL22] to adaptive/parallel settings likely needs new techniques. We explained the  $\geq 1/2 $ probability bound via Markov’s inequality (lines 231–233), specified when the Donsker–Varadhan principle applies, justified the initialization assumptions, and refined algorithmic details, including a comparison to applying the parallel scheme of [ACV24].

- **Practical and Implementation Aspects**: we formalized a space–adaptivity trade-off via a sliding-window strategy [SBE+24], and added a 2D Gaussian-mixture experiment: the sequential baseline [BCE+22] needs $2000$ iterations, while our parallel method achieves comparable accuracy in $300$. Additional experiments on non-log-concave targets are planned.

For the final version, we will:
- **Theory**:
Clarify the reason why we focus on large-FI lower bounds and highlight the difficulties of extending to small-FI in the final section; correct the small mistakes and add explanation for unclear bounds (Line 231-233).
- **Algorithm & presentation**:
Rewrite Section 3.1 to provide more intuition; revise Fig. 1 (diagonal update, cleaner visuals); add a high-level overview of the lower-bound proof; move formal complexity definitions to the appendix.
- **Practicality & experiments**:
Add a remark introducing a sliding-window scheme and the space–adaptivity trade-off; include 2D Gaussian-mixture results (sequential [BCE+22] $\approx 2000$ iterations vs. ours $\approx 300$).

---

### Decision · Program_Chairs · 2025-09-17

**Decision:**

Accept (poster)

**Comment:**

The authors consider the adaptive complexity of parallel sampling from non-log-concave distributions in relative Fisher information (a standard measure for this problem, similar to finding approximate stationary points in non-convex optimization). Using Picard iteration, they show improvement over sequential algorithms; they also prove a lower bound in a certain regime where parallelism does not improve.

This is a nice theoretical result with a new algorithm, and offers some insights of contrasts with parallel non-convex optimization. Reviewers found that the theoretical clarifications answered their questions. The one negative review (5zmn) cited that the lower bound was only for specific epsilon and the lack of experiments. Authors noted that in the parallel setting a more general lower bound requires new techniques, and conducted a 2-D experiment. Other reviewers agreed on the difficulty of proving lower bounds in general. The results seem adequate for a theory paper.

I remind the authors to incorporate the presentation of the algorithm and proof, as requested by reviewers.